# Decomposition of Concept-Level Rules in Visual Scenes

**Fan Shi**,* **Yuxuan Liang**,* **Xiaolei Chen**, **Haiyang Yu**, **Xu Li**, **Yi Zheng**, **Rui Zhu**,
**Xiangyang Xue**,† **Bin Li**†
Shanghai Key Laboratory of Intelligent Information Processing
College of Computer Science and Artificial Intelligence, Fudan University
`{fshi22,yxliang25,chenxl23}@m.fudan.edu.cn`
`hyyu20@fudan.edu.cn  {xu_li23,zhengy23,rzhu24}@m.fudan.edu.cn`
`{xyxue,libin}@fudan.edu.cn`

## Abstract

Human cognition is compositional, and one can parse a visual scene into independent concepts and the corresponding concept-changing rules. By contrast, many vision-language systems process images holistically, with limited support for explicit decomposition. Previous methods of decomposing concepts and rules often rely on hand-crafted inductive biases or human-designed priors. We introduce a Concept-Rule Decomposition (CRD) framework to decompose concept-level rules with Large Vision-Language Models (LVLMs), which explains visual input by leveraging LVLM-extracted concepts and the rules governing their variation. The proposed method operates in two stages: (1) a pretrained LVLM proposes visual concepts and concept values, which are employed to instantiate a space of concept rule functions that model concept changes and spatial distributions; (2) an iterative process to select a concise set of concepts that best account for the input according to the rule function. We evaluate CRD on an abstract visual reasoning benchmark, a spatial reasoning benchmark, and a real-world image caption dataset. Across both settings, our approach outperforms baseline models while improving interpretability by explicitly revealing underlying concepts and compositional rules, advancing explainable and generalizable visual reasoning.

## 1 Introduction

The compositionality of human cognition enables us to understand complex scenes by isolating independent concepts and their changes (Lake et al., 2017). A visual concept (Lake et al., 2015) is a high-level, interpretable visual property (i.e., meta-attribute) that describes specific semantic categories or abstract traits present (e.g., concept *Color* describes attributes such as *Blue*, *Red*, or *Green*). A Rule specifies the allowable pattern of how a concept's values vary across space (e.g., The *colors* of a rainbow are arranged from red to violet.) (Tenenbaum et al., 2011; Kemp & Tenenbaum, 2008). Many visual scenes involve such a decomposition into concepts and rules, for example, a matrix reasoning problem (Raven & Court, 1938) consists of human-defined meta-attributes and logical rules, while the entities in a physical video move under physical laws (McCloskey, 1983).

Early approaches decompose concepts and rules through hierarchical Bayesian inference, enabling the analysis of spatial arrangement rules in structured data (Kemp & Tenenbaum, 2008; Tenenbaum et al., 2011) as well as the compositional structure of handwriting characters (Lake et al., 2011; 2015). More recent approaches aim to disentangle the visual perception process from the high-level rule inference process by designing specialized modules (Zhang et al., 2021a), where structured generative priors, such as latent Gaussian processes (Shi et al., 2021) and algebraic reasoning backends (Zhang et al., 2021b), have been incorporated to provide effective inductive biases for abstract visual reasoning tasks. Another line of work enforces concept-specific latent functions to capture independent factors of variation (Shi et al., 2023), which is evaluated on three different visual scenes.

---

*Equal contribution
†Corresponding author

Prior works often rely on strong inductive biases or manually designed priors to extract interpretable structure. Although these inductive biases can lead to interpretable results, they also limit their adaptability to various visual scenes. Therefore, building frameworks that can automatically discover such compositional structure remains challenging. Motivated by these limitations, we leverage Large Vision-Language Models (LVLMs) as rich, data-driven priors for concept discovery and rule induction (Liu et al., 2023). LVLMs encode extensive world knowledge and fine-grained visual-linguistic correspondences (Wang et al., 2025b; Tong et al., 2024), and we could exploit these capacities to automatically perceive scene content, propose semantically meaningful concept candidates, and estimate their patch-wise values without manual attribute taxonomies or rule templates.

In this paper, we propose a **Concept-Rule Decomposition (CRD)** framework that leverages a pre-trained LVLM to decompose visual concepts and rules from visual inputs. First, CRD introduces a pre-trained LVLM to propose a *Visual Concept Set (VCS)* of the input, which contains a set of meta-attributes and the corresponding local attribute values. And we instantiate a *Concept Rule Function (CRF)* space to capture the change or spatial distribution of visual concepts. Second, given the CRF space grounded in the visual concepts, CRD performs an iterative sampling process to update the VCS that best accounts for the visual input. Through the sampling process, CRD converges on a set of visual concepts and associated rule functions that provide an interpretable explanation of the patterns observed in the visual data. To evaluate the proposed CRD, we construct a subset from VStar Bench (Wu & Xie, 2024), consisting of high-quality, open-domain real-world images paired with their corresponding meta-attributes.

The main contributions are summarized as follows:

- We propose CRD, a novel and general framework to decompose concept-level rules from visual inputs based on the perception and reasoning ability of LVLMs.

- We demonstrate that CRD can automatically extract and decompose concept-level rules from both natural and abstract images, improving the attribute extraction and spatial reasoning abilities of various LVLM baselines.

- We show that CRD enables LVLMs to surpass both traditional concept-rule decomposition methods and standard LVLM baselines on abstract visual reasoning tasks, highlighting its ability to decompose the visual concepts and rules.

## 2 RELATED WORKS

**Decomposition of Concepts and Rules.** Earlier approaches analyze spatial arrangement rules on grid-structured image panels via Hierarchical Bayesian inference (Tenenbaum et al., 2011; Kemp & Tenenbaum, 2008). In parallel, character modeling approaches decompose whole characters into strokes through predefined compositional processes and recombine them into new characters according to specific rules (Lake et al., 2011; 2015; Shi et al., 2025). Some methods leverage hand-crafted or learned feature representations to address abstract visual reasoning tasks (Lovett & Forbus, 2017; Little et al., 2012; Lovett et al., 2010). Recent works model concepts and rules more explicitly by combining latent encodings with probabilistic induction or adversarial learning (Pekar et al., 2020; Zhang et al., 2021a;b; Shi et al., 2024; 2026), while alternative approaches capture concepts and concept-changing rules through latent functions (Shi et al., 2021; 2023). Nevertheless, these methods still depend on auxiliary supervision or task-specific inductive biases, often involving human-injected knowledge like the specific form of rules.

**Large Vision-Language Models.** Large Vision-Language Models (LVLMs) have evolved significantly, starting with early models (Liu et al., 2023; Li et al., 2023; Chen et al., 2025) that connected pre-trained visual encoders (e.g., CLIP-based ViTs (Radford et al., 2021)) to language models for open-ended visual question answering. These models paved the way for later LVLMs (Liu et al., 2024; Chen et al., 2024b; Yin et al., 2025; Yu et al., 2025; Liang et al., 2025; 2026), which improved input image resolution and enhanced vision-language alignment. Notably, InternVL-3.5 (Wang et al., 2025a) and QwenVL2.5 (Bai et al., 2025) exemplify the latest advancements, with both introducing new training strategies and frameworks to optimize reasoning performance. Several recent approaches have sought to leverage the capabilities of LVLMs to perform visual reasoning (Tong et al., 2024; Wang et al., 2025b; Chen et al., 2024a). These innovations have closed the performance gap with proprietary models such as GPT-4o (Achiam et al., 2023), making open

LVLMs competitive in vision-language tasks and offering impressive improvements in multimodal reasoning. Despite substantial progress, current LVLMs remain limited in visual rule extraction and abstract visual reasoning. Trained largely for pattern recognition and caption-style objectives, they receive little supervision for inferring compositional rules or conducting systematic relational reasoning. Empirical analysis reports weak compositional understanding of concept-relation bindings (Anis et al., 2025) and persistent failures on abstract rule-induction tasks (Ahrabian et al., 2024). This motivates our approach, which harnesses LVLM priors while introducing a decomposition mechanism into interpretable concepts and rules, enabling more robust abstract visual reasoning.

## 3 METHOD

In this section, we describe how CRD decomposes visual inputs into visual concepts and rules. We first introduce the probabilistic definition of the visual concepts and rules in an overview, and then describe the two-stage process by which the model learns both components from raw data.

### 3.1 VISUAL CONCEPT SET

The set of candidate visual concepts contains a vast collection of possible concepts, often covering the majority of words in the vocabulary. As a result, the concept space is highly diverse, making exhaustive enumeration impractical. Let $[M] = \{1, \ldots, M\}$ denote the universe of all $M$ candidate visual concepts (we use their indices for convenience). Each element in $[M]$ corresponds to a primitive visual concept that could potentially be used to describe the input data. While $[M]$ defines the full concept vocabulary, only a very small portion of these concepts is related to the input.

**Definition 1** (Visual Concept Set). *Given the candidate concepts $[M]$, a Visual Concept Set (VCS) of size $K$ is a subset $G \subseteq [M]$ with cardinality $|G| = K$.*

According to Definition 1, CRD selects a subset from $G$ to form the VCS, which contains $K$ concepts that are actually relevant for explaining the given visual input $X$. For each visual concept $i \in [M]$, we assume that $p_i \in (0, 1)$ is the probability that $i$ is included in the VCS $G$, and the logit $\theta_i = \log \frac{p_i}{1-p_i}$. The probability distribution of $G$ given $\theta_i$ is (see Appendix A for the proof)

$$p_K(G \mid \theta) = \frac{1}{Z} \prod_{i \in G} e^{\theta_i}, \quad \text{where } Z = \sum_{\substack{S \subseteq [M] \\ |S| = K}} \prod_{j \in S} e^{\theta_j}. \tag{1}$$

A higher $\theta_i$ indicates a higher probability that concept $i$ is included in $G$. Furthermore, $\theta$ reflects the likelihood that the concept values are supported by specific rules, which establishes the connection between VCS and the rules. If the change pattern of a concept follows a certain rule, the concept is more likely to serve as a latent factor that appropriately explains the visual input. In this case, its corresponding score $\theta_i$ should be higher, resulting in a greater probability of the concept in $G$. In summary, $p_K(G \mid \theta)$ considers the underlying rules, biasing the selection toward concepts that exhibit clearer rules for structured decomposition of visual inputs. In the following section, we will introduce how CRD bridges the visual concepts and rules.

### 3.2 CONCEPT RULE FUNCTION

The rules in CRD capture the spatial distribution of concept values within an image. While concepts describe attributes such as color, the rules specify how these concept values are arranged and interact spatially. CRD learns rules by analyzing patch-wise concept values, capturing value changes and their spatial dependencies. As illustrated in Figure 1, CRD first splits the input image $X$ into $N$ non-overlapping patches $\{x_1, \ldots, x_N\}$, ordered in a raster-scan manner from left to right and top to bottom. A LVLM is employed to generate a set of concepts $G_{\text{rule}}$ according to $X$. For each concept $i \in G_{\text{rule}}$, the LVLM further extracts concept values of each patch. Let $p_n$ denote the spatial position of $x_n$, and $v_{i,n}$ is the observed value of concept $i$ on the $n$-th patch.

**Definition 2** (Concept Rule Function). *For a given input image $X$, the values of concept $i$ extracted by the LVLM are $\boldsymbol{v}_i = [v_{i,1}, \ldots, v_{i,N}]^\top$. The position vector of patches is $\boldsymbol{p} = [p_1, \ldots, p_N]^\top$. The Concept Rule Function (CRF) of concept $i$ is a mapping $f : \boldsymbol{p} \mapsto \boldsymbol{v}_i$. The function space $\mathcal{F}$ of the*

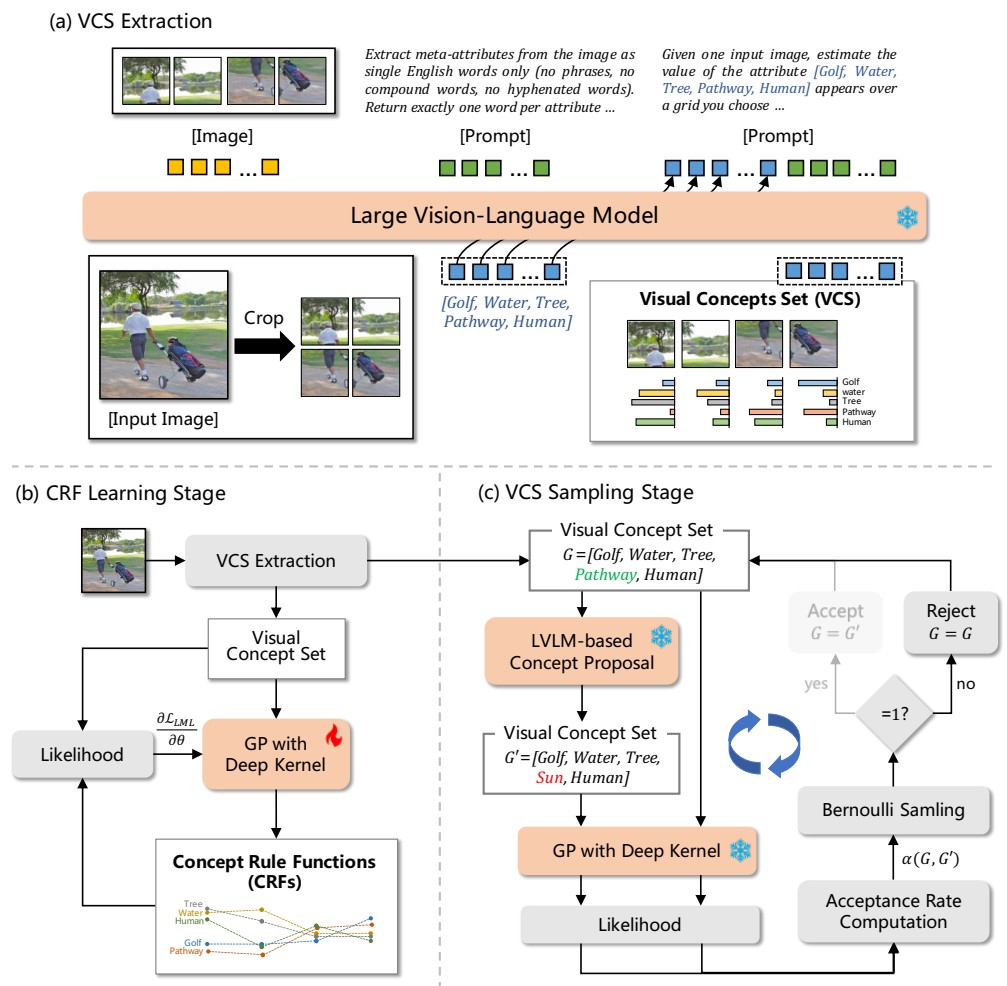

Figure 1: **Overview of the proposed CRD framework.** The process consists of two stages. In the CRF learning stage, visual concepts are extracted from images and a Gaussian process prior is used to construct the function space over CRFs. In the VCS sampling stage, a iterative sampling procedure is applied to generate concept subsets $G$ from the distribution $p_K(G \mid \theta)$.

*mapping is a Gaussian Process (GP) (Williams & Rasmussen, 1995) with a deep kernel:*

$$f \sim \mathcal{GP}(0, k_\phi(\cdot, \cdot)), \quad \text{where } k_\phi(p_i, p_j) = \exp\left(-\frac{1}{2}\big\|g_\phi(p_i) - g_\phi(p_j)\big\|_2^2\right), \quad 1 \le i, j \le N,$$

*and $g_\phi$ is a neural network that maps input positions into high-dimensional representations.*

The marginal likelihood of the concept values $\boldsymbol{v}_i$ is a Gaussian $p(\boldsymbol{v}_i \mid \boldsymbol{p}, \phi) = \mathcal{N}(\boldsymbol{v}_i; \boldsymbol{0}, \boldsymbol{K}_\phi)$ under the GP prior, where $\boldsymbol{K}_\phi \in \mathbb{R}^{N \times N}$ is the kernel matrix with entries $(\boldsymbol{K}_\phi)_{ij} = k_\phi(p_i, p_j)$. The logarithmic marginal likelihood is

$$\mathcal{L}_{\text{LML}}(\boldsymbol{p}, \boldsymbol{v}_i) = -\frac{1}{2}\boldsymbol{v}_i^\top \boldsymbol{K}_\phi^{-1}\boldsymbol{v}_i - \frac{1}{2}\log\det\left(\boldsymbol{K}_\phi\right) - \frac{N}{2}\log(2\pi). \tag{2}$$

Intuitively, a CRF characterizes how visual concepts vary across spatial positions of an image. Since the concepts with specific patterns are considered more related to the input, CRD defines $\theta_i$ through a CRF and the function space $\mathcal{F}$, i.e., $\theta_i = \mathcal{L}_{\text{LML}}(\boldsymbol{p}, \boldsymbol{v}_i)$.

### 3.3 LVLM-BASED TWO-STAGE LEARNING PROCESS

Based on Definitions 1 and 2, the main challenges that CRD needs to address are twofold:

1. how to construct the function space $\mathcal{F}$ to fit the rules on input images;

2. how to efficiently sample $G$ from the complex probability distribution $p_K(G \mid \theta)$.

To tackle both challenges, we propose a LVLM-based learning process shown in Figure 1. In the *CRF learning stage*, a LVLM is taken to extract visual concepts and learn an appropriate function space $\mathcal{F}$ over CRFs. In the second *VCS sampling stage*, according to the learned function space, we employ a sampling procedure like Metropolis-Hastings to generate $G$ from $p_K(G \mid \theta)$. In the following sections, we will provide a detailed description of this two-stage process.

### 3.3.1 CRF LEARNING STAGE

Given a batch of input images, we construct the patches and extract the patch positions and the concept values using a LVLM, forming the training set for the corresponding CRFs. The training set is denoted as $\mathcal{D} = \{(\boldsymbol{p}_i, \boldsymbol{v}_i)\}_{i=1}^N$, where $\boldsymbol{p}_i$ denotes the spatial positions of patches and $\boldsymbol{v}_i$ is the predicted concept values. To train the CRF, we minimize the negative logarithmic marginal likelihood defined in Equation 2 on the observed concept values. The parameters $\phi$ are optimized by gradient-based methods. We compute the gradient of the negative logarithmic marginal likelihood with respect to $\phi$. This gradient is then used in standard gradient descent or adaptive optimizers (e.g., Adam (Kingma & Ba, 2014)) to update $\phi$. By iteratively processing batches of images, computing the marginal likelihood, and performing gradient-based updates, the deep kernel parameters are learned such that the GP prior over CRFs captures the underlying rules of visual concepts across spatial positions.

### 3.3.2 VCS SAMPLING STAGE

Once the distribution $p_K(G \mid \theta)$ and the function space $\mathcal{F}$ have been defined, the second stage of our framework aims to obtain samples of $G$ that are consistent with both the probabilistic distribution. Direct sampling from $p_K(G \mid \theta)$ is computationally intractable due to the combinatorial size of the candidate concepts. To address this, we design a Metropolis-Hastings (MH) sampling procedure (Chib & Greenberg, 1995), denoted as *LVLM-MH*, which leverages the proposal distribution informed by a LVLM.

Starting from a current VCS $G$, we propose a new VCS $G' = G \setminus \{i\} \cup \{j\}$ by replacing the concept $i \in G$ with the candidate concept $j \in [M] \setminus G$. The transition probability of the concept replacement process is decomposed as

$$Q(G, G') = r(i \mid G)\, q(j \mid i, G), \tag{3}$$

where $r(i \mid G)$ denotes the probability of selecting concept $i$ of $G$ for replacement, and $q(j \mid i, G)$ denotes the proposal probability of selecting the candidate concept $j$ as its replacement. We consider the following design choices for $r(i \mid G)$. First, select $i \in G$ uniformly at random, i.e., $r(i \mid G) = 1/|G|$. Second, weight the selection by the inverse importance of concepts, i.e., $r(i \mid G) \propto e^{-\theta_i}$, such that less related concepts are more likely to be replaced. CRD determines the replaced concept by selecting $i \in G$ uniformly at random, which avoids dependence on the score $\theta$ assigned to each concept. This choice removes the necessity of evaluating $\theta$ over the entire set $G$ and brings a computational advantage. At the same time, uniform sampling guarantees an exploration of the concepts, ensuring that every candidate has equal opportunity to be selected.

After determining the replaced concept $i$, the distribution $q(j \mid i, G)$ is instantiated by the LVLM to propose a target concept $j$ from the candidate concepts $[M] \setminus G$. The distribution leverages the semantic prior captured by the LVLM, thereby assigning higher probability to concepts that are semantically or visually more consistent with the input image. This mechanism ensures that the sampling process is guided by high-level semantic knowledge, facilitating the discovery of more meaningful candidate concepts. To avoid degenerate cases where the LVLM assigns an extremely small probability to certain concepts, we impose a constraint on the logits before normalization. The output logits produced by the LVLM are clipped to ensure that no concept receives a probability arbitrarily close to zero.

With the transition probability $Q(G, G')$, the acceptance probability of LVLM-MH is computed by

$$\alpha(G, G') = \min \left\{ 1, \frac{p_K(G' \mid \theta)}{p_K(G \mid \theta)} \cdot \frac{r(j \mid G') \, q(i \mid j, G')}{r(i \mid G) \, q(j \mid i, G)} \right\},$$

$$= \min \left\{ 1, e^{\theta_j - \theta_i} \cdot \frac{q(i \mid j, G')}{q(j \mid i, G)} \right\}.$$

(4)

A Bernoulli variable is sampled with the acceptance rate $\alpha(G, G')$. The proposal $G \rightarrow G'$ will be executed if the sampled result is 1; otherwise, the current VCS is retained. For an input image, we run multiple iterations of the LVLM-MH sampler. At each iteration, a candidate concept is proposed and accepted or rejected with probability $\alpha(G, G')$. Over repeated iterations, the VCS converges to the target distribution $p_K(G \mid \theta)$, thus providing samples of concept sets consistent with the complex probability distribution of VCS.

## 4 EXPERIMENTS

### 4.1 BASELINES

To assess robustness and model-agnostic generality, we benchmark CRD against a diverse panel of large vision-language models that span families and parameter scales. Specifically, we include InternVL-3.5 in 4B/8B (Wang et al., 2025a), Qwen2.5-VL in 3B/7B (Bai et al., 2025) and DeepSeek-VL2-Tiny (Wu et al., 2024). We use publicly released checkpoints and official inference pipelines without any additional fine-tuning. To ensure comparability, we harmonize evaluation along three axes: (1) prompting standardized instruction templates for all baselines (detailed in Appendix C); (2) decoding deterministic generation with temperature 0; and (3) adopting each model's native image preprocessor (including any built-in high-resolution tiling or sub-image partition mechanisms). We also compare our method with six approaches based on deep learning in abstract visual reasoning tasks: PrAE (Zhang et al., 2021a), LGPP (Shi et al., 2021), and CLAP-NP (Shi et al., 2023), LEN (Zheng et al., 2019), ResNet+DRT (Zhang et al., 2019), and SRAN (Hu et al., 2021). These traditional deep learning methods are not general-purpose models in the sense of LVLMs. They are typically designed for a single dataset or a limited set of tasks, with architectures and training objectives specifically designed to capture dataset-dependent rules and concepts. In contrast, LVLMs provide a universal modeling paradigm, pretrained on large-scale multimodal corpora, and are capable of zero-shot or few-shot reasoning across diverse tasks.

### 4.2 DATASETS

**Meta-Attribute Extraction.** We derived VSB-MA by selecting all high-quality images from VS-tar Bench (Wu & Xie, 2024), aiming to establish a standardized set of meta-attributes for general visual scenes. Each image was first processed with a GPT-4o (Achiam et al., 2023) model using a carefully engineered prompt (detailed in the appendix) to automatically extract an initial pool of descriptive attributes. These raw attributes were then rigorously reviewed and cleaned by human annotators to ensure conceptual uniqueness of each attribute and clear inter-attribute distinctiveness. The cleaning process involved removing redundant or ambiguous descriptors, merging semantically overlapping terms under unified labels, and verifying consistency across similar scenes. As a result, we developed a curated and standardized meta-attribute set for each image, which serves as a high-quality reference for assessing the ability of models to extract visual features and rules in complex real-world scenarios.

**Abstract Visual Reasoning.** We evaluate our framework on RAVEN (Zhang et al., 2019) and I-RAVEN (Hu et al., 2021), two widely used datasets for abstract visual reasoning. Both datasets consist of seven image configurations with diverse layouts (e.g., single-object, inside-outside, and grid-based), where images are governed by compositional rules over attributes such as shape, size, and position. In addition, both datasets introduce noisy attributes (e.g., random rotations, colors, and object positions in grids), which increase the difficulty of learning concept-rule composition only from the raw data. In this task, we fill each candidate option into the question matrix and select the one with the highest CRD rule likelihood as the final answer.

**SpatialEval Benchmark.** We evaluate our framework on SpatialEval, a benchmark designed to probe spatial reasoning in LVLMs. SpatialEval spans four complementary tasks that cover core

Table 1: **Meta-attribute extraction performance on VSB-MA**. We compare the standard LVLM baselines with the corresponding CRD instantiations.

| Method | Avg. Sim. | Precision | Recall | F1 | AUPRC | ROC-AUC |
|---|---|---|---|---|---|---|
| DeepSeek-VL2-Tiny | 16.8 | 39.1 | 21.1 | 27.4 | 36.3 | 50.8 |
| DeepSeek-VL2-Tiny + CRD | **20.4** | **44.8** | **23.2** | **30.6** | **40.1** | **58.3** |
| Qwen2.5-VL-3B | 31.5 | 77.1 | 26.9 | 39.9 | 42.5 | 65.8 |
| Qwen2.5-VL-3B + CRD | 36.7 | 77.3 | 32.8 | 46.1 | 47.9 | 68.1 |
| Qwen2.5-VL-7B | 46.9 | 73.7 | 38.0 | 50.2 | 54.1 | 74.6 |
| Qwen2.5-VL-7B + CRD | **51.6** | **76.3** | **44.4** | **56.1** | **58.0** | **75.7** |
| InternVL-3.5-4B | 38.5 | 75.1 | 35.3 | 48.0 | 48.8 | 68.7 |
| InternVL-3.5-4B + CRD | 44.5 | 76.4 | 42.7 | 54.8 | 52.4 | 70.1 |
| InternVL-3.5-8B | 59.9 | 75.7 | 51.2 | 61.1 | 65.2 | 83.9 |
| InternVL-3.5-8B + CRD | **64.0** | **77.4** | **55.6** | **64.7** | **68.3** | **84.8** |
| Human | 77.4 | 84.7 | 74.6 | 79.3 | 79.0 | 87.7 |

facets of spatial intelligence, including spatial relationship understanding on maps, goal-directed navigation in mazes, position querying and counting in structured grids, and spatial reasoning grounded in real images with long dense captions. A key feature of the benchmark is that each problem provides both an image and a textual description that are individually sufficient to answer the question, enabling controlled evaluation under text-only, vision-only, and vision+text settings, and making it possible to analyze how models use or ignore visual evidence when rich textual cues are available. In our evaluation, CRD concatenates the extracted meta-attributes and patch scores as an additional prompt prefix to the original question and then obtains the model's response.

### 4.3 META-ATTRIBUTE EXTRACTION

**Metrics.** Let $\hat{\mathcal{A}}$ and $\mathcal{A}$ denote the sets of predicted attributes and ground-truth attributes, respectively. For any pair $(i, j) \in \hat{\mathcal{A}} \times \mathcal{A}$, the cosine similarity score $s(i, j) \in [-1, 1]$ is computed from the sentence-transformer (Reimers & Gurevych, 2019) embeddings of the two attributes. We obtain a one-to-one alignment matrix $M \subseteq \hat{\mathcal{A}} \times \mathcal{A}$ (e.g., through the Hungarian algorithm (Japrapto, 2010)) that maximizes total similarity. *Average Similarity* is the mean similarity over aligned pairs:

$$\text{AvgSim} = \frac{1}{|M|} \sum_{(i,j) \in M} s(i, j), \tag{5}$$

providing a graded measure of alignment quality across the matched concepts. We report Precision, Recall, F1, AUPRC, and ROC-AUC derived from the similarity scores, providing a comprehensive evaluation of the attribute extraction performance in terms of accuracy, coverage, and separability.

**Quantitative Results.** As shown in Table 1, CRD consistently boosts meta-attribute extraction performance for every model and scale evaluated. For each architecture, our method outperforms its base counterpart in all metrics. For example, the InternVL-3.5-4B's Average Similarity (AvgSim) rises from 38.5 to 44.5 and its F1 score from 48.0 to 54.8 after using CRD. We observe similar improvements for the larger InternVL-3.5-8B and Qwen2.5-VL-7B when optimized with CRD. Even smaller models benefit: Qwen2.5-VL-3B and DeepSeek-VL2-Tiny both show notable gains in all metrics after using our method. We also include the performance of human experts on the meta-attribute extraction task as a reference. Collectively, these results suggest that CRD enables the models to more fully tap into their vast pre-trained knowledge, substantially strengthening their ability to interpret complex visual rules and to extract the relevant attributes with higher fidelity. More experimental results are provided in Appendix D.

**Ablation Study.** Table 2 indicates the impact of removing each component from the acceptance probability $\alpha(G, G')$ in our VCS sampling stage. Removing any single component of the acceptance probability causes a notable drop in performance. This consistent decline confirms the effectiveness of our framework. The rule-based CRF score (the term $e^{\theta_j - \theta_i}$ derived from the Concept Rule Function) effectively guides the update of the concept set, while the LVLM proposal ratio (based on the LVLM's proposal probability) helps the large model thoroughly explore the concept space.

Table 2: **Ablation study on acceptance probability components in VCS sampling stage**. We analyze the impact of different components in the acceptance probability $\alpha(G, G')$ using InternVL-3.5-8B as the baseline model on VSB-MA dataset.

| Method | Avg. Sim. | Precision | Recall | F1 | AUPRC | ROC-AUC |
|---|---|---|---|---|---|---|
| InternVL-3.5-8B + CRD | **64.0** | **77.4** | **55.6** | **64.7** | **68.3** | **85.0** |
| w/o LVLM Proposal Ratio | 61.0 | 77.1 | 51.3 | 61.6 | 65.8 | 84.2 |
| w/o CRF Score Term | 59.2 | 74.6 | 49.8 | 57.4 | 64.8 | 84.1 |
| InternVL-3.5-8B | 59.9 | 75.7 | 51.2 | 61.1 | 65.2 | 84.5 |

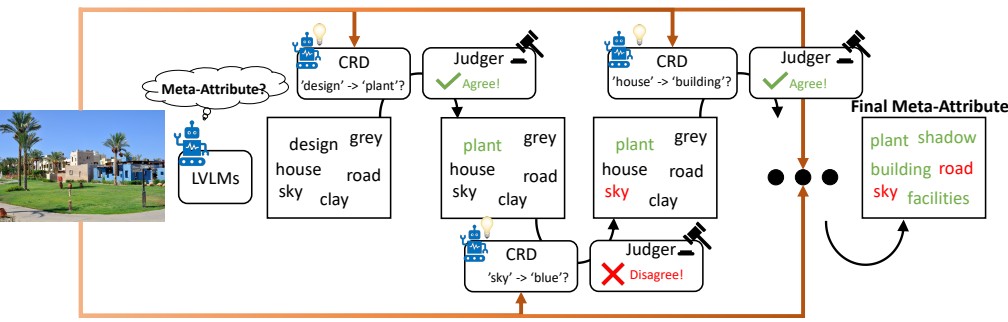

Figure 2: Qualitative case studies of meta-attribute extraction on the VSB-MA dataset.

Together, these components drive significant performance gains over the baseline. Furthermore, we compare the two ablated variants in detail. Removing the LVLM proposal ratio results in a performance decline (e.g., Avg Sim drops from 64.0 to 61.0), remaining above the baseline (59.9). This suggests that the CRF score term alone is a strong contributor to performance. The LVLM proposal ratio provides an additional benefit by allowing some proposals with lower CRF scores to be accepted. This encourages a broader exploration of the concept space and yields a higher overall performance. In contrast, removing the CRF score term causes a severe drop, and even below the baseline on most metrics. Without the CRF score to steer the sampling toward concept sets, unguided exploration can degrade the results. Appendix D provides more analyses of our framework.

**Qualitative Results.** As shown in Figure 2, given a natural image, the pretrained LVLM first proposes an initial pool of candidate meta-attributes. CRD then performs an iterative propose-judge-update loop. In each iteration, CRD suggests a refinement or replacement aimed at increasing semantic abstraction and rule consistency (e.g., replacing the vague design with the concept plant, or lifting the instance-level house to the category-level building). Proposals inconsistent with the meta-attribute definition (e.g., sky → blue, which collapses to a specific color instance) are rejected. Repeating this process converges to a concise, interpretable meta-attribute set that better aligns with the scene's underlying organization. More qualitative results can be found in Appendix E.

## 4.4 ABSTRACT VISUAL REASONING

In the abstract visual reasoning tasks, we compare our method with deep-learning methods, which require dataset-specific hyperparameter tuning (e.g., representation dimensions and number of rules) to adapt to different data. We instantiate CRD with InternVL-3.5, Qwen2.5-VL, and DeepSeek-VL2, and compare the resulting models against their original baselines as well as other representative LVLMs. In this setting, the context panel and all candidate images are concatenated into a single problem image, where the candidates are annotated from A to H. The complete prompt construction procedure is described in Appendix C. To solve abstract visual reasoning tasks with LVLM-CRD, the right-bottom panel of the problem matrix is iteratively replaced with each candidate image. For each replacement, the corresponding CRF score is computed to evaluate how well the completed matrix conforms to the learned rules. Finally, the candidate with the highest CRF score among all eight options is selected as the predicted answer for the model evaluation.

Table 3: **The performance on the abstract visual reasoning task**. We show the accuracy (%) of selecting answers on subsets of RAVEN/I-RAVEN. Both Qwen2.5-VL-3B and Qwen2.5-VL-7B achieve much higher accuracy on RAVEN but drops to near-random accuracy on I-RAVEN, despite the two datasets sharing the same context panels and differing only in the candidate set, raising the concern of data contamination. InternVL-CRD indicates the CRD instantiated with InternVL-3.5-8B, and Qwen-VL-CRD indicates the CRD instantiated with Qwen2.5-VL-7B. We use the 7B checkpoint of LLaVA-NeXT and the tiny version of DeepSeek-VL2.

| Models | RAVEN / I-RAVEN | | | | | | | |
|---|---|---|---|---|---|---|---|---|
| | Center | L-R | U-D | O-IC | O-IG | 2Grid | 3Grid | Average |
| PrAE | 14.5/22.6 | 7.1/21.2 | 11.1/26.5 | 7.1/16.9 | 9.5/24.4 | 13.1/21.4 | 11.1/18.9 | 10.5/21.7 |
| LGPP | 9.2/20.1 | 4.7/18.9 | 5.2/21.2 | 4.0/13.9 | 3.1/12.3 | 8.6/13.7 | 10.4/13.9 | 6.5/16.3 |
| CLAP-NP | 30.4/42.9 | 13.4/35.1 | 12.2/32.1 | 16.4/37.5 | 9.5/26.0 | 16.0/20.1 | 24.3/35.8 | 17.5/32.8 |
| ResNet+DRT | 14.1/13.2 | 11.9/13.4 | 12.8/12.1 | 13.6/12.1 | 13.1/13.3 | 16.8/12.4 | 16.1/12.8 | 14.1/12.8 |
| SRAN | 75.8/**89.6** | 31.0/67.6 | 33.2/70.9 | 39.3/75.7 | 68.0/52.2 | 66.9/38.6 | 79.3/32.2 | 56.2/61.0 |
| LEN | 69.3/15.3 | 74.5/14.6 | 74.2/15.5 | 72.8/12.8 | 77.6/15.7 | 65.0/15.1 | 73.5/16.1 | 72.4/15.0 |
| InternVL3-2B | 9.0/9.0 | 17.0/13.0 | 14.0/16.0 | 13.0/10.0 | 10.0/13.0 | 13.0/9.0 | 10.0/12.0 | 12.3/11.7 |
| +Instruct | 13.0/8.0 | 15.0/11.0 | 15.0/18.0 | 14.0/11.0 | 7.0/13.0 | 13.0/12.0 | 14.0/17.0 | 13.0/12.9 |
| InternVL3-8B | 16.0/15.0 | 6.0/14.0 | 12.0/13.0 | 11.0/13.0 | 17.0/14.0 | 13.0/11.0 | 6.0/17.0 | 11.6/13.9 |
| +Instruct | 12.0/16.0 | 12.0/10.0 | 11.0/11.0 | 12.0/11.0 | 16.0/15.0 | 10.0/10.0 | 8.0/18.0 | 11.6/13.0 |
| GPT-4o | 16.0/13.0 | 13.0/13.0 | 10.0/6.0 | 13.0/11.0 | 10.0/17.0 | 8.0/13.0 | 11.0/12.0 | 11.6/12.1 |
| LLaVA-NeXT | 14.0/14.0 | 13.0/13.0 | 20.0/17.0 | 10.0/9.0 | 10.0/12.0 | 9.0/15.0 | 13.0/12.0 | 12.7/13.1 |
| DeepSeek-VL2 | 18.0/17.0 | 22.0/15.0 | 11.0/11.0 | 21.0/7.0 | 20.0/15.0 | 17.0/11.0 | 10.0/11.0 | 17.0/12.4 |
| Qwen2.5-VL-3B | 68.0/17.0 | 54.0/17.0 | 33.0/11.0 | 71.0/11.0 | 54.0/15.0 | 55.0/5.0 | 44.0/8.0 | 54.1/12.0 |
| Qwen2.5-VL-7B | **78.0**/19.0 | 53.0/18.0 | 55.0/14.0 | 73.0/18.0 | 65.0/12.0 | 52.0/9.0 | 42.0/15.0 | 59.7/15.0 |
| InternVL-CRD | 22.0/24.0 | 19.0/29.0 | 31.0/37.0 | 33.0/44.0 | 47.0/49.0 | 33.0/20.0 | 36.0/32.0 | 31.6/33.6 |
| Qwen-VL-CRD | 77.0/81.0 | **97.0/93.0** | **95.0/95.0** | **87.0/89.0** | **98.0/98.0** | **84.0/80.0** | **88.0/89.0** | **89.4/89.3** |

**Quantitative Results.** Table 3 reports the experimental results on the classical abstract visual reasoning datasets RAVEN and I-RAVEN, where the models are evaluated by the accuracy of selecting the correct answer from eight candidates. The deep learning-based baselines achieve higher accuracy than most LVLMs, while LVLMs exhibit limited performance on both RAVEN and I-RAVEN. Open-source models such as InternVL3 and LLaVA-Next, as well as the closed-source GPT-4o, achieve accuracies close to random guessing (12.5%). An exception is Qwen2.5-VL, which reaches over 60% accuracy on RAVEN, substantially outperforming other LVLMs. Notably, when applied to Qwen, CRD already achieves performance that surpasses several of these task-specific models, especially on I-RAVEN. Recent work (Jiang et al., 2025) has also observed similar anomalies. Qwen2.5-VL performs well on the final answer selection task, which is the original RAVEN task, but struggles on simpler intermediate reasoning subtasks, leading the authors to suspect potential data contamination. We observe a similar discrepancy in our experiments. The proposed method mostly outperforms both deep learning-based baselines and LVLMs on these benchmarks, demonstrating its advantage in abstract reasoning. We hypothesize that the poor reasoning performance of LVLMs stems from their lack of explicit problem decomposition ability. Unlike CRD, which transforms the problems into a structured concept-rule learning process, LVLMs tend to rely on holistic pattern matching, making it difficult to capture the underlying abstract rules behind problems. Appendix E provides additional qualitative results and analysis.

## 4.5 SPATIALEVAL BENCHMARK

Table 4 reports the results on SpatialEval Benchmark (Wang et al., 2024) to evaluate four spatial reasoning abilities: SpatialMap, MazeNav, SpatialGrid, and SpatialReal. We compare the original LVLMs with two variants of our method. CRD-meta uses only the extracted meta-attributes as additional prompts. While CRD-full further incorporates patch-level scores. Across both Qwen2.5-VL-3B and Qwen2.5-VL-7B, CRD-meta already provides clear gains over the base models, and CRD-full yields further improvements, particularly in SpatialReal and MazeNav. These results show that both types of CRD-derived signals contribute meaningful guidance to spatial reasoning tasks.

Table 4: **Performance on SpatialEval Benchmark.** SpatialEval includes four spatial reasoning tasks (SpatialMap, MazeNav, SpatialGrid, SpatialReal). CRD-meta denotes that only the meta-attributes extracted by CRD are used as prompts to guide the LVLM's responses. CRD-full further incorporates both the meta-attributes and the patch scores as additional prompts for the LVLMs.

| Model | SpatialMap | MazeNav | SpatialGrid | SpatialReal | Overall |
|---|---|---|---|---|---|
| SpatialVLM | 48.87 | 22.60 | **87.03** | 34.81 | 50.27 |
| Qwen2.5-VL-3B | 50.93 | 27.40 | 83.33 | 91.85 | 54.00 |
| Qwen2.5-VL-3B+CRD-meta | 50.93 | **29.07** | 85.20 | **96.30** | 56.27 |
| Qwen2.5-VL-3B+CRD-full | **51.67** | 29.00 | 86.40 | **96.30** | **56.87** |
| Qwen2.5-VL-7B | 63.00 | 28.93 | 85.60 | 91.11 | 60.11 |
| Qwen2.5-VL-7B+CRD-meta | 64.20 | 29.13 | 87.93 | 95.56 | 61.45 |
| Qwen2.5-VL-7B+CRD-full | **65.26** | **33.33** | **88.46** | **97.04** | **63.37** |

Table 5: **Efficiency analysis on the VSB-MA benchmark.** We report the average latency per token (ms), computational cost (TFLOPs), GPU memory usage (GiB), KV-Cache size (MB), and input token count for different models with and without CRD. We also analyze the impact of patch granularity on computational efficiency by varying the number of patches (2×2, 3×3, 4×4) for InternVL-3.5-8B on the VSB-MA benchmark.

| Model | Latency Per Token | TFLOPs | GPU Memory | KV-Cache | Input Token |
|---|---|---|---|---|---|
| InternVL-3.5-8B | **77.3** | **39.02** | **17.02** | **1287.0** | **2288** |
| +CRD-2×2 | 234.3 | 42.93 | 17.46 | 1405.7 | 2499 |
| +CRD-3×3 | 236.1 | 43.23 | 17.67 | 1414.7 | 2515 |
| +CRD-4×4 | 239.5 | 43.71 | 17.95 | 1429.3 | 2541 |
| Qwen2.5-VL-7B | **39.7** | **6.47** | **16.49** | **171.1** | **447** |
| +CRD-2×2 | 190.1 | 8.91 | 17.32 | 234.7 | 613 |
| DeepSeek-VL2-Tiny | **59.6** | **1.06** | **11.82** | **114.8** | **1224** |
| +CRD-2×2 | 264.1 | 1.16 | 12.53 | 124.9 | 1332 |

## 4.6 EFFICIENCY ANALYSIS

As shown in Table 5, we summarize the inference efficiency of different models with and without CRD. The GP-based CRF and sampling procedure do introduce additional computation compared to direct LVLM inference. However, this extra cost comes with more accurate concept-rule decomposition, which in turn leads to significantly improved performance on the evaluated reasoning tasks. In our setting, the number of patches per image is kept small. The default CRD configuration uses a $2 \times 2$ spatial grid (i.e., N = 4 patches), so the theoretical $O(N^3)$ cost of GP inference remains negligible in practice. We provide a complexity/latency table reporting latency and memory usage under different patch counts, showing that runtime grows modestly and remains practical within the patch range we consider. Scalable GP variants such as SKI (Wilson & Nickisch, 2015) are mainly beneficial when N is very large (e.g., thousands of patches); since our method operates in a low-N regime by design, approximate inference is not considered here.

## 5 CONCLUSION

We presented CRD, a model-agnostic framework that leverages pretrained LVLMs to propose visual concepts, which is employed to learn rule functions that capture how these concepts vary and organize spatially. Followed by an iterative sampling process, CRD selects a proper set of visual concepts and concept-level rules for the input. Across natural-image and abstract visual reasoning evaluations, CRD improves performance over LVLM baselines, demonstrating its ability to decompose concepts and rules. By minimizing hand-crafted inductive biases and harnessing data-driven priors, CRD offers a general route to explainable, compositional visual understanding. In future work, we plan to extend the framework to temporal and broader settings and to explore richer rule spaces and sampling strategies.

ACKNOWLEDGMENTS

This work was supported by the National Natural Science Foundation of China (No.62176060) and the Program for Professor of Special Appointment (Eastern Scholar) at Shanghai Institutions of Higher Learning.

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

# A PROOFS

## A.1 PROOFS OF EQUATION 1

We consider the probability distribution of a visual concept set $G$, where each concept $i \in [M]$ is included in $G$ with probability $p_i \in (0,1)$. Denote $|G| = K$ and define $\theta_i = \log \frac{p_i}{1-p_i}$. The goal is to derive

$$p_K(G \mid \theta) = \frac{1}{Z} \prod_{i \in G} e^{\theta_i}, \quad \text{where } Z = \sum_{\substack{S \subseteq [M] \\ |S|=K}} \prod_{j \in S} e^{\theta_j}.$$

**Proof.** Without the constraint $|G| = K$, the probability of $G$ is a product of independent Bernoulli distributions:

$$p(G) = \prod_{i \in G} p_i \prod_{i \notin G} (1 - p_i).$$

Imposing the constraint $|G| = K$, the conditional probability is

$$p_K(G) = \frac{\prod_{i \in G} p_i \prod_{i \notin G} (1 - p_i)}{\sum_{S \subseteq [M], |S|=K} \prod_{j \in S} p_j \prod_{j \notin S} (1 - p_j)}.$$

Factor out the term $C(p) = \prod_{t=1}^{M} (1 - p_t)$:

$$\sum_{\substack{S \subseteq [M] \\ |S|=K}} \prod_{j \in S} p_j \prod_{j \notin S} (1 - p_j) = C(p) \cdot \sum_{\substack{S \subseteq [M] \\ |S|=K}} \prod_{j \in S} \frac{p_j}{1 - p_j} = C(p) \cdot \sum_{\substack{S \subseteq [M] \\ |S|=K}} \prod_{j \in S} e^{\theta_j},$$

$$\prod_{i \in G} p_i \prod_{i \notin G} (1 - p_i) = C(p) \cdot \prod_{i \in G} \frac{p_i}{1 - p_i} = C(p) \cdot \prod_{i \in G} e^{\theta_i}.$$

Then, we obtain

$$p_K(G \mid \theta) = \frac{\prod_{i \in G} e^{\theta_i}}{\sum_{S \subseteq [M], |S|=K} \prod_{j \in S} e^{\theta_j}}.$$

□

# B DATASETS

## B.1 ABSTRACT VISUAL REASONING

We evaluate the methods on two commonly used abstract visual reasoning datasets: RAVEN (Zhang et al., 2019) and I-RAVEN (Hu et al., 2021). Both datasets contain seven distinct image configurations, as illustrated in Figure 3. The *Center* configuration contains a single central object, while *L-R* and *U-D* consist of two objects arranged horizontally or vertically. *O-IC* and *O-IG* adopt inside-outside layouts, and *2Grid* and *3Grid* contain 2×2 and 3×3 object grids, respectively. While most configurations involve rules applied to a single component, O-IG and grid-based configurations additionally introduce rules defined over object grids.

Each configuration involves abstract rules that govern visual attributes, including four main categories: **Constant**, **Progress**, **Arithmetic**, and **Distribution Three**.

1. **Constant**: the attribute keeps unchanged in rows;
2. **Progress**: the attribute increases or decreases with the same stride in rows;
3. **Arithmetic**: the attribute of the third image is computed from the attributes of the first two images via specific arithmetic operations (e.g., addition and subtraction operations);
4. **Distribution Three**: the attributes in rows are three fixed values in different orders.

In addition, both datasets contain noise attributes, which are randomly sampled from the feasible set (e.g., object rotation in non-grid settings, or rotation, color, and grid position in grid-based settings).

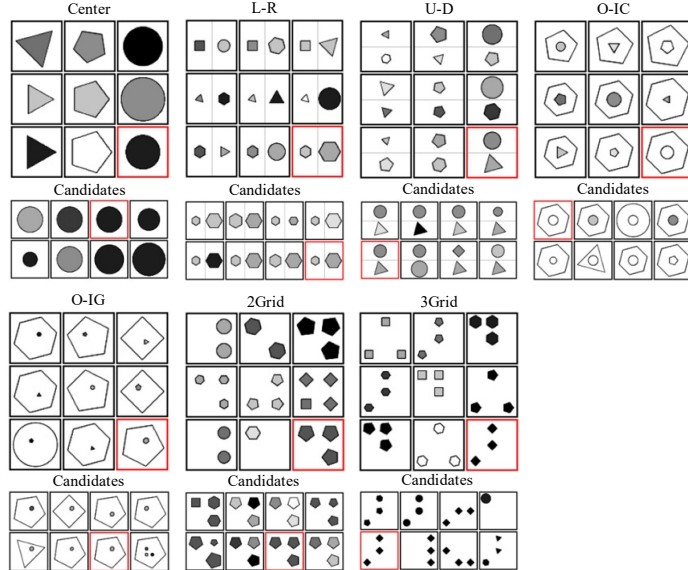

Figure 3: **Visualization of the RAVEN dataset.** The seven configurations on RAVEN/I-RAVEN. The images with red borders are correct answers that fix the rules defined in the panels.

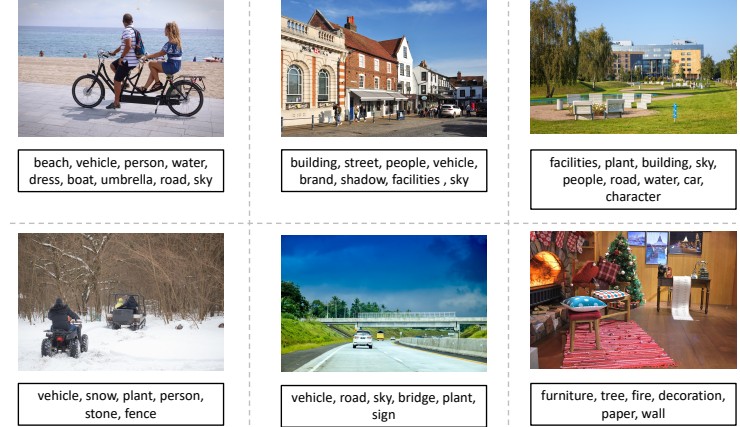

Figure 4: **VSB-MA dataset examples.** Sample images from the VSB-MA dataset derived from VStar Bench, showing diverse visual scenes with their corresponding meta-attributes for concept-rule decomposition evaluation.

## B.2   META-ATTRIBUTE EXTRACTION

As shown in Figure 4, VSB-MA is a full set of all high-quality natural images sampled from VStar Bench, created to provide a standardized reference for meta-attribute-based concept-level rules evaluation. For each image, we first elicit an initial attribute pool with GPT-4V using a tightly engineered prompt (specification included below). We then conduct rigorous human curation to enforce conceptual uniqueness and inter-attribute separability: redundant or ambiguous descriptors are removed, semantically overlapping items are merged under a unified label, and cross-scene consistency is verified. Meta-attributes are defined as high-level, interpretable properties (e.g., vehicle, building, furniture, plant, shadow, road, sky), rather than instance-level categories (e.g., car, chair) or low-level appearances (e.g., specific colors like blue). The resulting per-image meta-attribute sets form a clean, taxonomy-consistent target that spans diverse indoor/outdoor scenes and supports reliable assessment of both concept extraction and rule identification in complex real-world imagery.

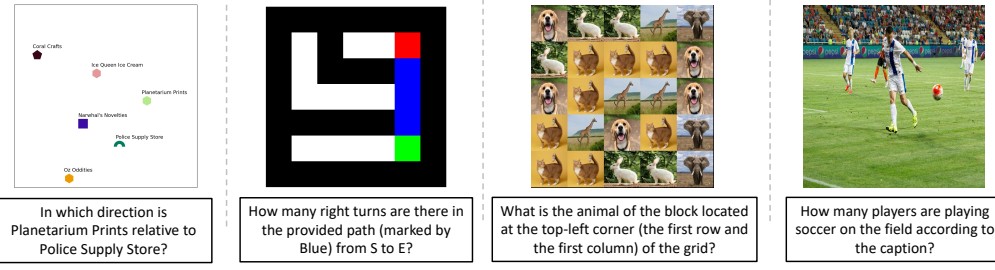

Figure 5: **Visualization of the SpatialEval Benchmark.** SpatialEval is a comprehensive benchmark for evaluating spatial intelligence in LLMs and VLMs across four key dimensions: spatial relationships, positional understanding, object counting, and navigation.

### B.3 SPATIALEVAL BENCHMARK

We also evaluate our framework on SpatialEval, a benchmark designed to assess spatial reasoning in both language models and vision language models. As shown in Figure 5, SpatialEval comprises four complementary tasks, including Spatial Map, Maze Nav, Spatial Grid, and Spatial Real, which collectively cover spatial relation understanding on maps, goal-oriented navigation in mazes, position querying and counting in structured grids, and spatial reasoning grounded in real images. The benchmark contains multiple-choice questions for each task, totaling 4,635 questions, and each example is constructed so that the image alone and the text alone are each sufficient to determine the answer. This design supports controlled evaluation under text-only, vision-only, and combined vision plus text settings, and it enables fine-grained analysis of whether a model genuinely uses visual evidence when strong textual cues are also available.

## C IMPLEMENTATION DETAILS

### C.1 PROMPT CONSTRUCTION FOR ABSTRACT VISUAL REASONING

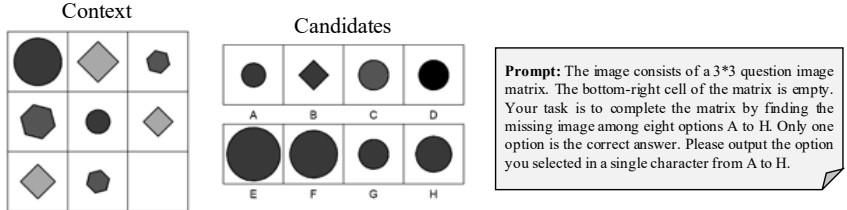

Figure 6: The prompt construction process of LVLMs on RAVEN/I-RAVEN.

For the LVLMs that support visual inputs, including InternVL3 (Wang et al., 2025a), LLaVA-NeXT (Liu et al., 2024), Qwen2.5-VL (Bai et al., 2025), and GPT-4o (Achiam et al., 2023), each RAVEN problem is transformed into a single composite image. The left part shows the $3 \times 3$ matrix with the bottom-right cell missing, while the left part contains the eight candidate images labeled from *A* to *H*. This composite image is then provided to the LVLMs, along with a prompt that describes the task and specifies the required output format. An illustration of this construction process is shown in Figure 6.

### C.2 LVLMS BASELINES AND COMMON PROTOCOL

We evaluate CRD on a diverse panel of pretrained LVLMs spanning families and parameter scales: InternVL-3.5 (4B/8B), Qwen-VL-2.5 (3B/7B), and DeepSeek-VL2-Tiny. We use the publicly released checkpoints (or official API for GPT-4o) and their native inference pipelines. As reported

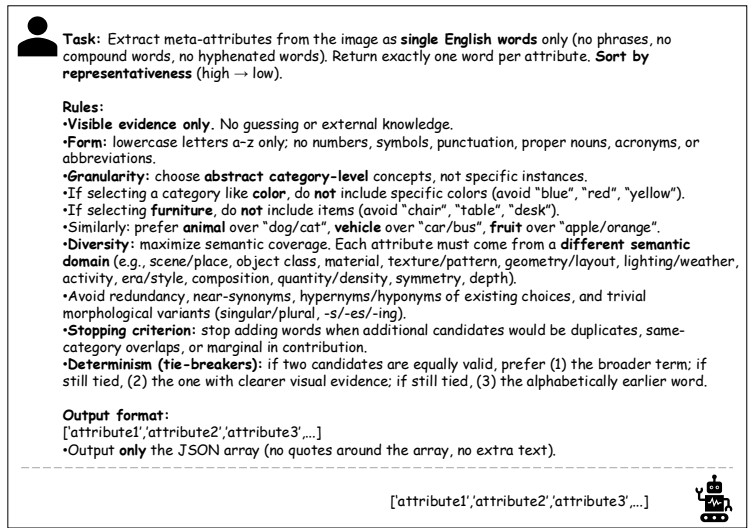

Figure 7: Prompt-1. Initialize the meta-attributes list for the given image.

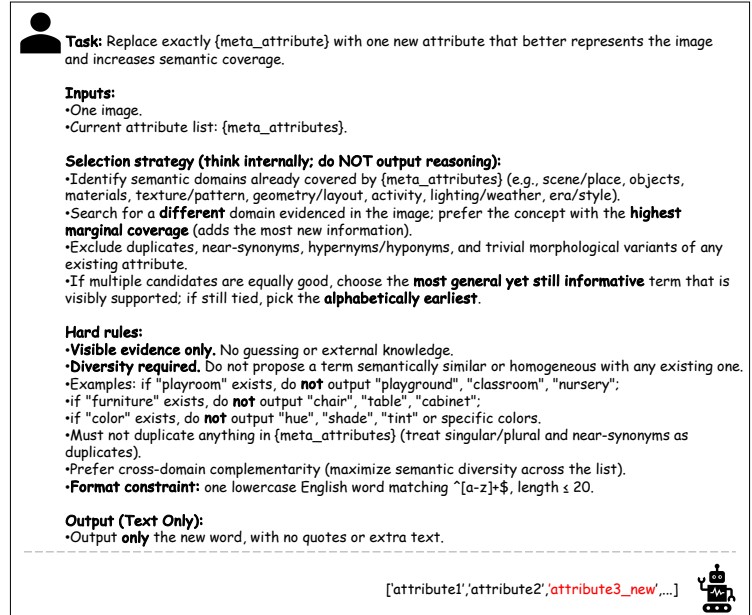

Figure 8: Prompt-2. Replace exactly the meta-attribute with one new attribute that better represents the image and increases semantic coverage.

by their authors, the InternVL models employ high-performance dynamic sub-image partitioning to accommodate ultra-high-resolution inputs, pair these with larger vision encoders, and align-tune the vision backbone directly to the LLM backbone. The Qwen-VL-2.5 models accept high-resolution images without explicit tiling, leveraging a strong vision encoder and high-quality training for broad coverage. DeepSeek-VL2 follows an architecture conceptually similar to LLaVA (vision encoder + projection + LLM), but reports stronger empirical performance across many multimodal tasks. We keep each model's native image preprocessor (including any built-in tiling/partition logic or resize policy) to avoid confounding changes.

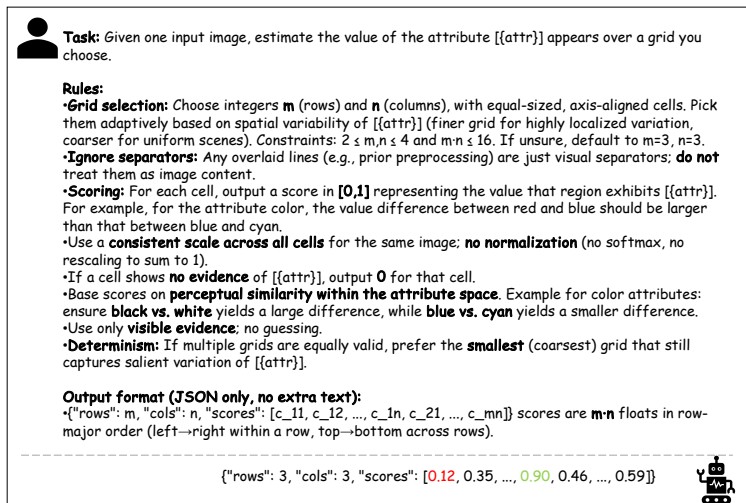

Figure 9: The prompt-3. Given one input image, estimate the value of an attribute that appears over a grid.

To ensure comparability, we standardize prompting, decoding, and preprocessing across baselines. We use fixed instruction templates for all models (the exact prompts are shown in the Figure 7 8 9), and adopt deterministic generation with $temperature = 0$ and $do\_sample = False$ (no nucleus/top-k sampling). For logit clipping, we retain only candidate tokens whose probabilities are at least $10\%$ of the maximum logit. For the replacement selection step, we substitute tokens only when their replacement probability exceeds $0.9$, and we perform 5 replacement-selection iterations in total. Unless otherwise stated, we make no architectural changes, apply no additional fine-tuning, and perform no extra training on any baseline; all results are obtained with the official inference code and default checkpoints under the above common protocol.

## D  SUPPLEMENTARY EXPERIMENT

### D.1  VSB-MA BENCHMARK WITH HUMAN ANNOTATION

Table 6: **Performance on VSB-MA benchmark with human annotation.** We compare the performance of different LVLMs with and without CRD on the only human-annotated VSB-MA dataset. The metrics include Average Similarity, Precision, Recall, F1 Score, AUPRC, and ROC-AUC.

| Model | Avg. Sim. | Precision | Recall | F1 | AUPRC | ROC-AUC |
|---|---|---|---|---|---|---|
| InternVL-3.5-4B | 50.36 | 66.72 | 40.21 | 50.18 | 68.33 | 78.10 |
| +CRD | 58.30 | **82.15** | 47.76 | 60.40 | 72.06 | 82.85 |
| InternVL-3.5-8B | 58.56 | 72.31 | 51.39 | 60.08 | 72.71 | 89.51 |
| +CRD | **69.22** | 80.02 | **60.81** | **69.10** | **79.13** | **92.17** |
| Qwen2.5-VL-3B | 43.81 | 60.37 | 33.31 | 42.93 | 65.63 | 76.47 |
| +CRD | 49.40 | 73.92 | 37.68 | 49.92 | 69.58 | 82.24 |
| Qwen2.5-VL-7B | 55.27 | 64.03 | 47.21 | 54.35 | 69.87 | 82.74 |
| +CRD | **61.86** | **76.19** | **52.50** | **62.16** | **73.67** | **87.31** |
| DeepSeek-VL2-Tiny | 25.72 | 31.00 | 23.37 | 26.65 | 55.04 | 66.19 |
| +CRD | **29.01** | **40.25** | **26.16** | **31.71** | **57.12** | **69.64** |

Table 6 summarizes model performance on the human-annotated VSB-MA benchmark. We report six metrics that reflect both similarity-based and classification-based accuracy. Across all LVLM backbones, applying CRD leads to clear improvements, with gains observed in Average Similarity, Precision, Recall, F1, AUPRC, and ROC-AUC. Larger models such as InternVL-3.5-8B and

Qwen2.5-VL-7B benefit the most, but even smaller models like DeepSeek-VL2-Tiny show consistent enhancements. These results indicate that CRD provides reliable and transferable benefits when evaluated against human-labeled ground truth.

# E    CASE STUDY

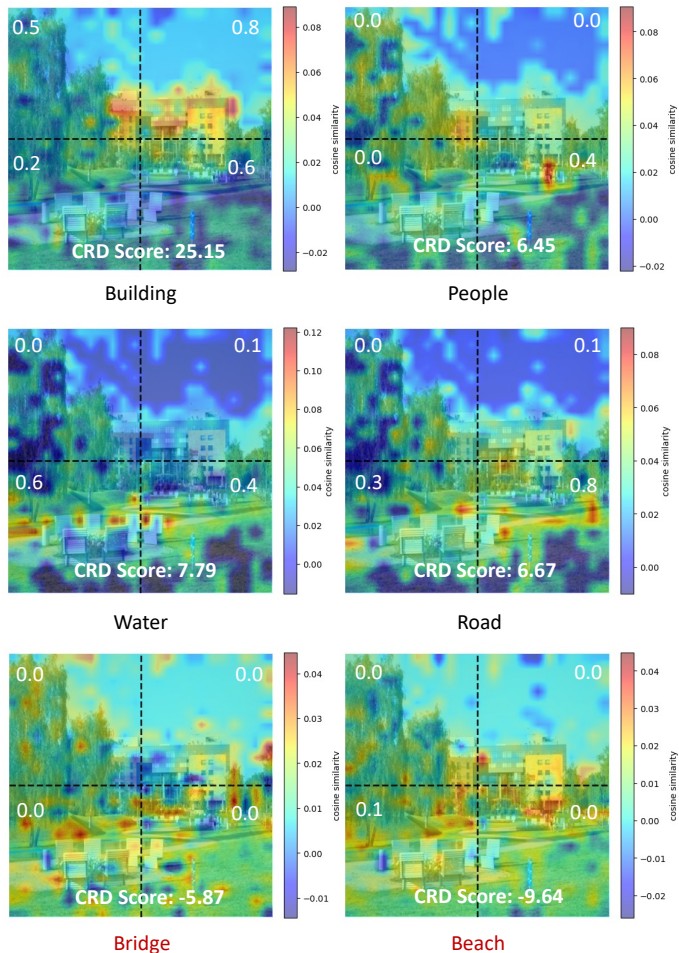

Figure 10: **Additional case studies on VSB-MA dataset.** Meta-attributes present in the image are marked in black, while those not present are marked in red.

## E.1    CASE STUDY OF META-ATTRIBUTE EXTRACTION

Figure 10 presents a case study from the VSB-MA dataset, where we extract multiple meta-attributes from a single image. Meta-attributes that are truly present in the image are annotated in black, whereas those absent from the scene are annotated in red. For each meta-attribute, we visualize a heatmap that reflects its spatial association with the image regions. This heatmap is obtained by computing the cosine similarity between the image-patch embeddings and the meta-attribute embedding, and then mapping the similarity scores back to the corresponding image tokens. We further annotate each image patch with the score assigned by CRD for that attribute, along with the attribute's final CRD Score. The spatial patterns align well with the semantics of the scene: for example, *People* receives its highest patch-level score exactly in the regions where humans appear, and the dominant meta-attribute *Building* obtains a strong positive CRD Score. Attributes that are absent in the image (e.g., *Bridge* and *Beach*) receive negative scores, which makes them more likely to be replaced during the iterative rule refinement process.

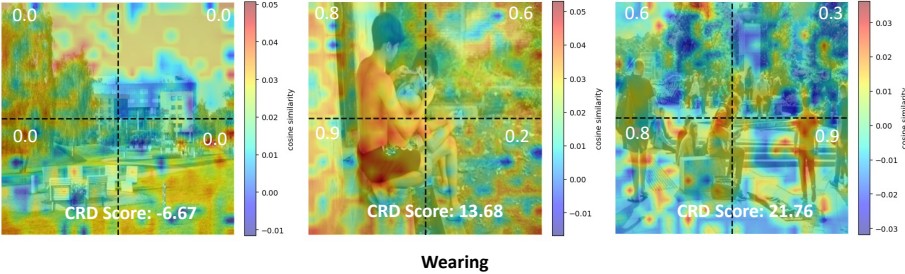

**Wearing**

Figure 11: **Additional case study with meta-attribute *Wearing*.** The visualization shows the effect of meta-attribute *Wearing* in different images.

Figure 11 further visualizes the non-object meta-attribute *Wearing* across different images. In the image on the right, the people are wearing different styles of clothing, and in the middle image, the person is not wearing a top, which is also considered a form of *wearing*, while the image on the left does not show any clear clothing-related features. CRD learns rules that assign appropriate scores to different patches. These results indicate that CRD can robustly extract interpretable rules for both object-centric and non-object meta-attributes.

### E.2 CASE STUDY OF ABSTRACT VISUAL REASONING

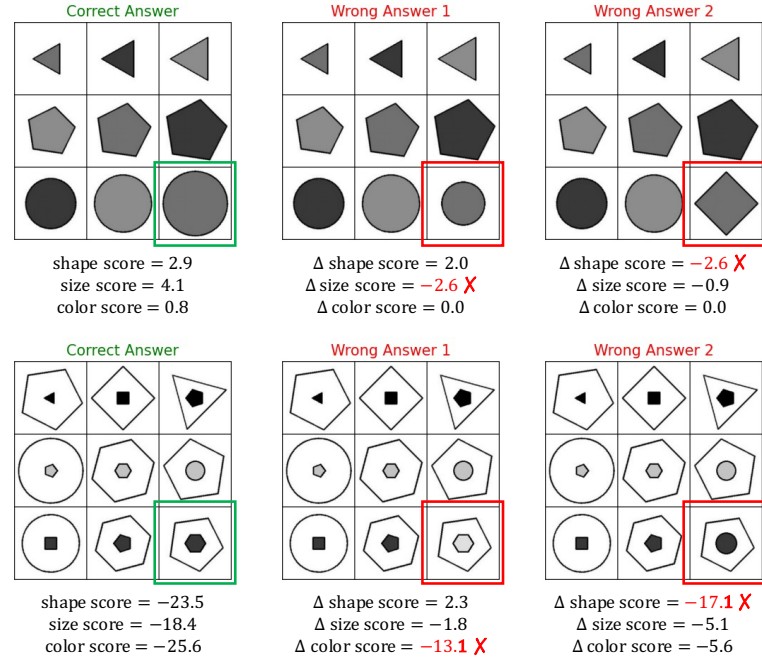

Figure 12: **Visualization of abstract visual reasoning examples.** Each row shows a reasoning problem where the correct answer produces coherent attribute patterns, while incorrect answers lead to notable score drops in the corresponding meta-attributes.

As shown in Figure 12, each row presents an example of an abstract reasoning problem. When the correct answer is filled in, the image matrix shows a coherent transformation pattern, such as object size increasing with a consistent step. When an incorrect answer is inserted, the scores of the corresponding meta attributes change accordingly. For example, if an incorrect size is provided, the size score drops by 2.6. This demonstrates that the meta attribute scores computed by CRD reliably capture whether the attribute values across the image patches follow the expected pattern.

## F    HUMAN ANNOTATION ETHICS AND PROCEDURES

In conducting our human annotation process, we followed established ethical practices to ensure that the data collection was responsible, transparent, and respectful of annotators. All annotators were recruited voluntarily and were informed of the purpose of the study, the nature of the tasks involved, and the expected time commitment. Before participating, each annotator provided explicit consent and was reminded that they could withdraw from the task at any time without consequence. To protect privacy, no personal identifying information was collected, and all annotations were stored in a secure environment accessible only to the research team. Annotators were given clear guidelines on how to perform the tasks and were compensated fairly in accordance with local standards. We also implemented quality control procedures, including multi-round checks and consistency reviews, to maintain the reliability of the collected labels. These measures aim to ensure that the dataset creation process is ethical, reproducible, and aligned with responsible research practices.

## G    LLM USAGE STATEMENT

We use Large Language Models (LLMs) as auxiliary tools during the preparation of this paper. The usage is limited to correcting grammatical issues, improving readability, and polishing the presentation. In addition, we use Large Vision-Language Models (LVLMs) as a critical component of the proposed framework. They are employed to extract visual concepts from visual input and to iteratively update the visual concept set for further refinement.

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
