# OpenReview forum: "Decomposition of Concept-Level Rules in Visual Scenes"
_ICLR.cc/2026/Conference — ICLR 2026 Poster_

### Official Review · Reviewer_3kLB · 2025-10-31

**Soundness:** 3
**Presentation:** 3
**Contribution:** 2
**Rating:** 4
**Confidence:** 4

**Summary:**

This paper introduces CRD, a two-stage framework designed to decompose visual scenes into a compact set of human-interpretable concepts and the rules governing their spatial variation. In Stage 1, a pretrained LVLM is used to propose a Visual Concept Set (VCS) and estimate patch-wise concept values. These values parameterize a Concept Rule Function (CRF), modeled as a deep-kernel Gaussian Process, whose log marginal likelihood serves as a per-concept interpretability score. In Stage 2, the framework samples an optimized size-K VCS using an LVLM-guided Metropolis–Hastings (MH) replacement kernel that evaluates proposals via an LVLM-based acceptance ratio. Experiments on two settings: (i) a curated VSB-Sub meta-attribute benchmark of 100 real images, and (ii) RAVEN/I-RAVEN abstract reasoning datasets, demonstrate consistent improvements over vanilla LVLMs for attribute extraction.

**Strengths:**

- **Originality**: This paper introduces an integration of Large Vision-Language Models (LVLMs) with Gaussian Process–based rule modeling, forming a bridge between probabilistic reasoning and LVLM-driven concept extraction;
- **Quality**: Methodologically solid: the two-stage inference process (concept proposal + CRF learning) is well-motivated and mathematically grounded in GP theory; Results improve meta-attribute extraction across diverse LVLMs and sizes (Table 1).
- **Clarity**: The paper is well-organized, with clear definitions for components such as Visual Concept Set (VCS) and Concept Rule Function (CRF). Figures (especially Figure 1) effectively convey the pipeline and improve understanding of the model’s decomposition process.
- **Significance**: Advances explainable visual reasoning; Demonstrates that a lightweight, post-hoc decomposition can boost both perception-level extraction and abstract reasoning without retraining the LVLM.

**Weaknesses:**

- **Missing baselines**: Authors compare CRD to PrAE, LGPP, and CLAP-NP on the RAVEN datasets, but several important baselines are missing — including LEN, ResNet + DRT, CoPINet, MRNet, and SRAN, which are standard on RAVEN and I-RAVEN. These models specifically target abstract visual reasoning and compositional rule induction, providing a fairer comparison for CRD’s intended goals.
- **Data scale**: The natural-image study uses a 100-image subset (VSB-Sub).  There is no clear indication on how the subset was selected. While authors mention the dataset was curated, this is small; thus generalization to varied scenes/domains remains uncertain.
- **Limited statistical reporting**: Tables omit variance, confidence intervals, or multiple runs, so performance differences might not be statistically significant.
- **Scalability & efficiency**: GP inference is O(N^3) in the number of patches. The paper doesn’t quantify runtime vs. patch count or images, nor compare with scalable kernels/inducing points. A complexity/latency table would help.
- **Reproducibility details**: Some choices (e.g., logit clipping of LVLM proposals, replacement selection policy trade-offs, MH schedule/temperature, number of steps) are described but not specified with hyperparameters or seeds.
- **Human study**: The paper involves human annotators for dataset curation and evaluation but provides no discussion on ethical considerations, recruitment, consent, or reproducibility of the human annotation process.

**Questions:**

1. How does inference time scale with the number of patches or image size? Could you provide empirical runtime or memory profiles?
2. Have you tried scalable GPs (e.g., SVGP or SKI) to see if performance holds under approximate inference?
3. How was the 100-image VSB-Sub subset selected? Was any stratification or randomization used to avoid bias?
4. How sensitive is CRD to the choice of pretrained LVLM?
5. In Table 3, the CRD framework shows modest performance cоmpared to LVLM baselines. Could the authors elaborate on this observation? How can the performance improve on these benchmarks using CRD?

---

> ### Author Response · Authors · 2025-11-27
> **Response to Reviewer 3kLB (Part 1)**
>
> We greatly appreciate Reviewer’s insightful comments. Below, we provide detailed responses to the raised points. If you have any further questions, please feel free to raise them, and we will be glad to address them.
>
> **Q1.**
>
> > **Missing baselines**: Authors compare CRD to PrAE, LGPP, and CLAP-NP on the RAVEN datasets, but several important baselines are missing — including LEN, ResNet + DRT, CoPINet, MRNet, and SRAN, which are standard on RAVEN and I-RAVEN. These models specifically target abstract visual reasoning and compositional rule induction, providing a fairer comparison for CRD’s intended goals.
>
> We appreciate the reviewer’s suggestion regarding additional baselines. The models mentioned (LEN, ResNet+DRT, CoPINet, MRNet, SRAN) are task-specific architectures designed exclusively for abstract visual reasoning on RAVEN-style datasets, but CRD is not tailored to this task. CRD is a more general module that operates on top of LVLMs without task-specific training. As shown in Tables 1 and 2, we therefore report CRD results using two representative LVLM backbones on both RAVEN and I-RAVEN. Notably, when applied to Qwen, CRD already achieves performance that surpasses several of these task-specific models, especially on I-RAVEN.
>
> **Table 1. Performance comparison on the RAVEN dataset.**
>
>
> $$
> \\begin{array}{l|c|c|c|c|c|c|c|c}
> \\hline
> \\textbf{Model} & \\textbf{Average} & \\textbf{CS} & \\textbf{LR} & \\textbf{UD} & \\textbf{OIC} & \\textbf{OIG} & \\textbf{2Grid} & \\textbf{3Grid} \\\\
> \\hline
> ResNet+DRT            & 14.1 & 14.1 & 11.9 & 12.8 & 13.6 & 13.1 & 16.8 & 16.1 \\\\
> SRAN                  & 56.1 & 75.8 & 31.0 & 33.2 & 39.3 & 68.0 & 66.9 & 79.3 \\\\
> LEN                   & 72.4 & 69.3 & 74.5 & 74.2 & 72.8 & 77.6 & 65.0 & 73.5 \\\\
> \\hline
> InternVL3.5-8B        & 11.6 & 16.0 & 6.0  & 12.0 & 11.0 & 17.0 & 13.0 & 6.0  \\\\
> InternVL3.5-8B+CRD    & 31.6 & 22.0 & 19.0 & 31.0 & 33.0 & 47.0 & 33.0 & 36.0 \\\\
> \\hline
> Qwen2.5VL-7B          & 59.7 & \\textbf{78.0} & 53.0 & 55.0 & 73.0 & 65.0 & 52.0 & 42.0 \\\\
> Qwen2.5VL-7B+CRD      & \\textbf{89.4} & 77.0 & \\textbf{97.0} & \\textbf{95.0} & \\textbf{87.0} & \\textbf{98.0} & \\textbf{84.0} & \\textbf{88.0} \\\\
> \\hline
> \\end{array}
> $$
>
>
>
> **Table 2. Performance comparison on the I-RAVEN dataset.**
>
>
> $$
> \\begin{array}{l|c|c|c|c|c|c|c|c}
> \\hline
> \\textbf{Model} & \\textbf{Average} & \\textbf{CS} & \\textbf{LR} & \\textbf{UD} & \\textbf{OIC} & \\textbf{OIG} & \\textbf{2Grid} & \\textbf{3Grid} \\\\
> \\hline
> ResNet+DRT            & 12.8 & 13.2 & 13.4 & 12.1 & 12.1 & 13.3 & 12.4 & 12.8 \\\\
> SRAN                  & 61.0 & \\textbf{89.6} & 67.6 & 70.9 & 75.7 & 52.2 & 38.6 & 32.2 \\\\
> LEN                   & 15.0 & 15.3 & 14.6 & 15.5 & 12.8 & 15.7 & 15.1 & 16.1 \\\\
> \\hline
> InternVL3.5-8B      & 13.9 & 15.0 & 14.0 & 13.0 & 13.0 & 14.0 & 11.0 & 17.0 \\\\
> InternVL3.5-8B+CRD  & 33.6 & 24.0 & 29.0 & 37.0 & 44.0 & 49.0 & 20.0 & 32.0 \\\\
> \\hline
> Qwen2.5VL-7B        & 15.0 & 19.0 & 18.0 & 14.0 & 18.0 & 12.0 & 9.0  & 15.0 \\\\
> Qwen2.5VL-7B+CRD    & \\textbf{89.3} & 81.0 & \\textbf{93.0} & \\textbf{95.0} & \\textbf{89.0} & \\textbf{98.0} & \\textbf{80.0} & \\textbf{89.0} \\\\
> \\hline
> \\end{array}
> $$

---

> > ### Author Response · Authors · 2025-11-27
> > **Response to Reviewer 3kLB (Part 2)**
> >
> > **Q2.**
> >
> > > **Data scale**: The natural-image study uses a 100-image subset (VSB-Sub). There is no clear indication on how the subset was selected. While authors mention the dataset was curated, this is small; thus generalization to varied scenes/domains remains uncertain.
> > >
> > > How was the 100-image VSB-Sub subset selected? Was any stratification or randomization used to avoid bias?
> >
> > We thank the reviewer for this helpful comment. In the original submission, we used a randomly selected 100-image subset to accelerate validation and analysis. In the revised manuscript, we report results on the full V* dataset to address concerns about sample size and selection, and the conclusions remain consistent with those from the subset. Furthermore, to better demonstrate the  generalization of CRD, we additionally evaluate our method on the SpatialEval benchmark, which covers a broader range of spatial and relational reasoning scenarios. As shown in Tables 3 and 4, the performance on both V* and SpatialEval [1] supports the extensibility of CRD across more diverse scenes and domains.
> >
> > **Table 3. Results on the SpatialEval benchmark.**  SpatialEval includes four spatial reasoning tasks (SpatialMap, MazeNav, SpatialGrid, SpatialReal). CRD_meta denotes only the meta-attributes extracted by CRD are used as prompts to guide the LVLM’s responses. CRD_full further incorporates both the meta-attributes and the patch scores as additional prompts for the LVLMs.
> >
> >
> > $$
> > \\begin{array}{l|c|c|c|c|c}
> > \\hline
> > \\textbf{Model} & \\textbf{SpatialMap} & \\textbf{MazeNav} & \\textbf{SpatialGrid} & \\textbf{SpatialReal} & \\textbf{Overall} \\\\
> > \\hline
> > SpatialVLM                        & 48.87 & 22.60 & \\textbf{87.03} & 34.81 & 50.27 \\\\
> > Qwen2.5VL-3B                     & 50.93 & 27.40 & 83.33 & 91.85 & 54.00 \\\\
> > Qwen2.5VL-3B+CRD\\_meta          & 50.93 & \\textbf{29.07} & 85.20 & \\textbf{96.30} & 56.27 \\\\
> > Qwen2.5VL-3B+CRD\\_full          & \\textbf{51.67} & 29.00 & 86.40 & \\textbf{96.30} & \\textbf{56.87} \\\\
> > \\hline
> > Qwen2.5VL-7B                     & 63.00 & 28.93 & 85.60 & 91.11 & 60.11 \\\\
> > Qwen2.5VL-7B+CRD\\_meta          & 64.20 & 29.13 & 87.93 & 95.56 & 61.45 \\\\
> > Qwen2.5VL-7B+CRD\\_full          & \\textbf{65.26} & \\textbf{33.33} & \\textbf{88.46} & \\textbf{97.04} & \\textbf{63.37} \\\\
> > \\hline
> > \\end{array}
> > $$
> >
> >
> >
> > **Table 4. Performance comparison on the full dataset.**
> >
> > $$
> > \\begin{array}{l|c|c|c|c|c|c}
> > \\hline
> > \\textbf{Model} & \\textbf{Average Similarity} & \\textbf{Precision} & \\textbf{Recall} & \\textbf{F1} & \\textbf{AUPRC} & \\textbf{ROC-AUC} \\\\
> > \\hline
> >  InternVL3.5-4B        & 38.5 & 75.1 & 35.3 & 48.03 & 48.8 & 68.7 \\\\
> >  +CRD                  & 44.5 & 76.4 & 42.7 & 54.78 & 52.4 & 70.1 \\\\
> >  InternVL3.5-8B        & 59.9 & 75.7 & 51.2 & 61.08 & 65.2 & 83.9 \\\\
> >  +CRD                  & \\textbf{64.0} & \\textbf{77.4} & \\textbf{55.6} & \\textbf{64.71} & \\textbf{68.3} & \\textbf{84.8} \\\\
> > \\hline
> >  Qwen2.5VL-3B         & 31.5 & 77.1 & 26.9 & 39.88 & 42.5 & 65.8 \\\\
> >  +CRD                  & 36.7 & 77.3 & 32.8 & 46.06 & 47.9 & 68.1 \\\\
> > Qwen2.5VL-7B         & 46.9 & 73.7 & 38.0 & 50.15 & 54.1 & 74.6 \\\\
> > +CRD                  & \\textbf{51.6} & \\textbf{76.3} & \\textbf{44.4} & \\textbf{56.13} & \\textbf{58.0} & \\textbf{75.7} \\\\
> > \\hline
> >  DeepSeekVL2-Tiny     & 16.8 & 39.1 & 21.1 & 27.41 & 36.3 & 50.8 \\\\
> >   +CRD                  & \\textbf{20.4} & \\textbf{44.8} & \\textbf{23.2} & \\textbf{30.57} & \\textbf{40.1} & \\textbf{58.3} \\\\
> > \\hline
> > \\end{array}
> > $$
> >
> >
> > [1] Wang, Jiayu, et al. "Is a picture worth a thousand words? delving into spatial reasoning for vision language models." NeurIPS 2024.
> >
> > **Q3.**
> >
> > > **Limited statistical reporting**: Tables omit variance, confidence intervals, or multiple runs, so performance differences might not be statistically significant.
> >
> > We thank the reviewer for raising this concern. In all our experiments, we configure the LVLM with `temperature` set to 0 and `do_sample` set to false, so the model produces identical outputs for the same input.

---

> > > ### Author Response · Authors · 2025-11-27
> > > **Response to Reviewer 3kLB (Part 3)**
> > >
> > > **Q4.**
> > >
> > > > **Scalability & efficiency**: GP inference is O(N^3) in the number of patches. The paper doesn’t quantify runtime vs. patch count or images, nor compare with scalable kernels/inducing points. A complexity/latency table would help.
> > > >
> > > > How does inference time scale with the number of patches or image size? Could you provide empirical runtime or memory profiles?
> > > >
> > > > Have you tried scalable GPs (e.g., SVGP or SKI) to see if performance holds under approximate inference?
> > >
> > > We thank the reviewer for this insightful question on scalability and efficiency. In our setting, the number of patches per image is kept small: the default CRD configuration uses a 2 $\\times$ 2 spatial grid (i.e., N = 4 patches), so the theoretical O(N^3) cost of GP inference remains negligible in practice. We provide a complexity/latency table (Table 5) reporting inference time and memory usage under different patch counts, showing that runtime grows modestly and remains practical within the patch range we consider. Scalable GP variants such as SVGP or SKI are mainly beneficial when N is very large (e.g., thousands of patches); since our method operates in a low-N regime by design, approximate inference is not considered here.
> > >
> > > **Table 5. Efficiency Analysis Table with different number of patches (N).**
> > >
> > >
> > > $$
> > > \\begin{array}{l|c|c|c|c|c|c}
> > > \\hline
> > > \\textbf{Model} & \\textbf{Latency Per Token (ms)} & \\textbf{Total Time (s)} & \\textbf{TFLOPs} & \\textbf{GPU Memory (GiB)} & \\textbf{KV-Cache (MB)} & \\textbf{Input Token} \\\\
> > > \\hline
> > >   InternVL3.5-8B   & 77.3  & 557.099 & 39.02 & 17.02 & 1287.0 & 2288 \\\\
> > >   +CRD-2\*2         & 234.3 & 968.844 & 42.93 & 17.46 & 1405.7 & 2499 \\\\
> > >   +CRD-3\*3         & 236.1 & 973.122 & 43.23 & 17.67 & 1414.7 & 2515 \\\\
> > >   +CRD-4\*4         & 239.5 & 980.236 & 43.71 & 17.95 & 1429.3 & 2541 \\\\
> > > \\hline
> > > \\end{array}
> > > $$
> > >
> > >
> > > **Q5.**
> > >
> > > > **Reproducibility details**: Some choices (e.g., logit clipping of LVLM proposals, replacement selection policy trade-offs, MH schedule/temperature, number of steps) are described but not specified with hyperparameters or seeds.
> > >
> > > We thank the reviewer for pointing out the missing hyperparameter details. We provide full reproducibility information for all components of our pipeline. For logit clipping, we retain only candidate tokens whose probabilities are at least 10% of the maximum logit. For the replacement selection, we substitute tokens only when their replacement probability exceeds 0.9. Throughout all experiments, the LVLM temperature is fixed to 0 and do_sample is set to false, ensuring deterministic outputs. We perform 5 replacement-selection iterations. These details have been added to the revision.
> > >
> > > **Q6.**
> > >
> > > > **Human study**: The paper involves human annotators for dataset curation and evaluation but provides no discussion on ethical considerations, recruitment, consent, or reproducibility of the human annotation process.
> > >
> > > We thank the reviewer for this important reminder. In the revised manuscript, we have added a detailed description in the appendix covering the ethical considerations of our human annotation process, including annotator recruitment, consent procedures, and guidelines followed to ensure responsible data collection and reproducibility.
> > >
> > > In conducting our human annotation process, we followed established ethical practices to ensure that the data collection was responsible, transparent, and respectful of annotators. All annotators were recruited voluntarily and were informed of the purpose of the study, the nature of the tasks involved, and the expected time commitment. Before participating, each annotator provided explicit consent and was reminded that they could withdraw from the task at any time without consequence. To protect privacy, no personal identifying information was collected, and all annotations were stored in a secure environment accessible only to the research team. Annotators were given clear guidelines on how to perform the tasks and were compensated fairly in accordance with local standards. We also implemented quality control procedures, including multi-round checks and consistency reviews, to maintain the reliability of the collected labels. These measures aim to ensure that the dataset creation process is ethical, reproducible, and aligned with responsible research practices.

---

> > > > ### Author Response · Authors · 2025-11-27
> > > > **Response to Reviewer 3kLB (Part 4)**
> > > >
> > > > **Q7.**
> > > >
> > > > > How sensitive is CRD to the choice of pretrained LVLM?
> > > >
> > > > We thank the reviewer for raising this question. As shown in Table 4, different pretrained LVLMs exhibit varying initial abilities, and this naturally affects the downstream rule learning and sampling performance. Our abstract reasoning experiments (Tables 1 and 2) also demonstrate that while CRD consistently improves over each backbone, the magnitude of improvement depends on the LVLM’s underlying capacity. Since CRD is designed to decompose the concepts and visual rules through LVLMs, a model with no meaningful concept extraction ability would offer limited knowledge for CRD to leverage. Across all tested baselines, we observe that LVLM possesses a reasonable conceptual understanding ability. For instance, Qwen shows particularly large gains on I-RAVEN after applying CRD. These results indicate that CRD reliably unlocks more of the LVLM’s potential, with stronger backbones yielding even larger benefits.
> > > >
> > > > **Q8.**
> > > >
> > > > > In Table 3, the CRD framework shows modest performance cоmpared to LVLM baselines. Could the authors elaborate on this observation? How can the performance improve on these benchmarks using CRD?
> > > >
> > > > In the original version of our manuscript, Table 3 reported our results only for InternVL with CRD, where CRD produced improvements over the InternVL baseline but did not surpass QwenVL. This is expected, as the effectiveness of CRD is related to the capability of the underlying backbone model. In the revised manuscript, we further include results for QwenVL with CRD (shown in Tables 1 and 2), which more clearly illustrate the potential of our framework. Although QwenVL suffers a substantial performance drop when moving from RAVEN to I-RAVEN, applying CRD recovers a large portion of this gap and yields clear gains over the raw QwenVL backbone. These results indicate that, even on strong LVLM baselines, CRD can stably enhance performance and amplify the underlying  capabilities on these benchmarks.

---

### Official Review · Reviewer_8JaY · 2025-11-01

**Soundness:** 2
**Presentation:** 2
**Contribution:** 2
**Rating:** 4
**Confidence:** 3

**Summary:**

The paper presents CRD (Concept Rule Decomposition), a two-stage framework that leverages large vision–language models (LVLMs) to automatically discover and decompose concept-level rules in visual scenes. In the first stage, a pre-trained LVLM proposes visual concepts and their corresponding concept values. In the second stage, the method iteratively selects a compact subset of these concepts that best explains the visual input, effectively uncovering interpretable concept–rule relationships within the image.

**Strengths:**

Decomposing visual reasoning into concept–rule pairs is an interesting idea, and leveraging LVLMs for this purpose is novel. The work clearly builds upon prior research in compositional reasoning while introducing a more data-driven and scalable formulation. Moreover, the experimental results demonstrate that integrating CRD consistently improves the performance of LVLMs.

**Weaknesses:**

1. Unclear writing and motivation. The motivation behind the work is not clearly articulated. While the authors discuss how human visual perception is compositional, it remains unclear why such a decomposition framework is needed in the context of LVLMs or what specific applications it enables. Additionally, the stated contributions are somewhat confusing. One claim suggests that CRD improves the visual representation capabilities of LVLMs, while another claims that CRD outperforms standard LVLMs. Since CRD itself builds upon LVLMs, it is unclear whether the comparison is against the same backbone or a new model.
2. Unproven interpretability claims. The paper repeatedly emphasizes interpretability, but no clear experiments or metrics are provided to validate this claim. Qualitative evidence (e.g., visualization of learned concepts or rules) or human evaluation would strengthen this aspect significantly.
3. Lack of qualitative and quantitative baselines. The paper provides limited qualitative examples, making it difficult to understand what CRD produces in practice. Moreover, several relevant baselines such as SpatialVLM and VisuoThink, which also address compositional or structured visual reasoning, are discussed but not directly compared experimentally. Including such comparisons would help contextualize the claimed improvements.
4. Minor: The font size in Figure 1 is small and difficult to read; increasing it would improve presentation clarity.

**Questions:**

See weakness.

---

> ### Author Response · Authors · 2025-11-27
> **Response to Reviewer 8JaY (Part 1)**
>
> We sincerely thank reviewer for the thoughtful and constructive comments and welcome any further suggestions. We address the concerns in detail below.
>
> **Q1.**
>
> > Unproven interpretability claims. The paper repeatedly emphasizes interpretability, but no clear experiments or metrics are provided to validate this claim. Qualitative evidence (e.g., visualization of learned concepts or rules) or human evaluation would strengthen this aspect significantly.
>
> We appreciate the reviewer’s comment on interpretability and have strengthened this aspect in the revised manuscript. First, we provide quantitative evaluations on the full V* dataset, including human-expert assessments of meta-attribute extraction and additional comparisons using purely human-labeled ground truth. The results shown  in the Tables 1 and 2 demonstrate that CRD consistently improves performance.
>
>
>
> **Table 1. Performance comparison on the full dataset.** Human represents a comparison between human responses and human-gut annotations.
>
> $$
> \\begin{array}{l|c|c|c|c|c|c}
> \\hline
> \\textbf{Model} & \\textbf{Average Similarity} & \\textbf{Precision} & \\textbf{Recall} & \\textbf{F1} & \\textbf{AUPRC} & \\textbf{ROC-AUC} \\\\
> \\hline
>  InternVL3.5-4B        & 38.5 & 75.1 & 35.3 & 48.03 & 48.8 & 68.7 \\\\
>  +CRD                  & 44.5 & 76.4 & 42.7 & 54.78 & 52.4 & 70.1 \\\\
>  InternVL3.5-8B        & 59.9 & 75.7 & 51.2 & 61.08 & 65.2 & 83.9 \\\\
>  +CRD                  & \\textbf{64.0} & \\textbf{77.4} & \\textbf{55.6} & \\textbf{64.71} & \\textbf{68.3} & \\textbf{84.8} \\\\
> \\hline
> SpatialVLM   & 25.9 & 60.1 & 18.1 & 27.82 & 39.9 & 63.6 \\\\
>  Qwen2.5VL-3B         & 31.5 & 77.1 & 26.9 & 39.88 & 42.5 & 65.8 \\\\
>  +CRD                  & 36.7 & 77.3 & 32.8 & 46.06 & 47.9 & 68.1 \\\\
> Qwen2.5VL-7B         & 46.9 & 73.7 & 38.0 & 50.15 & 54.1 & 74.6 \\\\
> +CRD                  & \\textbf{51.6} & \\textbf{76.3} & \\textbf{44.4} & \\textbf{56.13} & \\textbf{58.0} & \\textbf{75.7} \\\\
> \\hline
>  DeepSeekVL2-Tiny     & 16.8 & 39.1 & 21.1 & 27.41 & 36.3 & 50.8 \\\\
>   +CRD                  & \\textbf{20.4} & \\textbf{44.8} & \\textbf{23.2} & \\textbf{30.57} & \\textbf{40.1} & \\textbf{58.3} \\\\
> \\hline
>  Human                 & 77.43 & 84.67 & 74.63 & 79.33 & 79.03 & 87.73 \\\\
> \\hline
> \\end{array}
> $$
>
>
>
>
> **Table 2. Performance comparison on the full dataset with human annotation.**
>
>
> $$
> \\begin{array}{l|c|c|c|c|c|c}
> \\hline
> \\textbf{Model} & \\textbf{Average Similarity \\S} & \\textbf{Precision \\S} & \\textbf{Recall \\S} & \\textbf{F1 Score \\S} & \\textbf{AUPRC \\S} & \\textbf{ROC-AUC \\S} \\\\
> \\hline
> InternVL3.5-4B        & 50.36 & 66.72 & 40.21 & 50.18 & 68.33 & 78.10 \\\\
> +CRD                  & 58.30 & \\textbf{82.15} & 47.76 & 60.40 & 72.06 & 82.85 \\\\
> InternVL3.5-8B        & 58.56 & 72.31 & 51.39 & 60.08 & 72.71 & 89.51 \\\\
> +CRD                  & \\textbf{69.22} & 80.02 & \\textbf{60.81} & \\textbf{69.10} & \\textbf{79.13} & \\textbf{92.17} \\\\
> \\hline
> Qwen2.5-VL-3B         & 43.81 & 60.37 & 33.31 & 42.93 & 65.63 & 76.47 \\\\
> +CRD                  & 49.40 & 73.92 & 37.68 & 49.92 & 69.58 & 82.24 \\\\
> Qwen2.5-VL-7B         & 55.27 & 64.03 & 47.21 & 54.35 & 69.87 & 82.74 \\\\
> +CRD                  & \\textbf{61.86} & \\textbf{76.19} & \\textbf{52.50} & \\textbf{62.16}  & \\textbf{73.67} & \\textbf{87.31} \\\\
> \\hline
> DeepSeek-VL2-Tiny     & 25.72 & 31.00 & 23.37 & 26.65 & 55.04 & 66.19 \\\\
> +CRD                  & \\textbf{29.01} & \\textbf{40.25} & \\textbf{26.16} & \\textbf{31.71} & \\textbf{57.12} & \\textbf{69.64} \\\\
> \\hline
> \\end{array}
> $$
>
>
> Second, we have added qualitative visualizations in the Appendix E. These include relevance heatmaps showing how each extracted meta-attribute corresponds to specific image regions, as well as examples of the rules and associated scores inferred by CRD across different image patches. Together, these quantitative and qualitative results provide clearer evidence of the interpretability and effectiveness of the proposed framework.

---

> > ### Author Response · Authors · 2025-11-27
> > **Response to Reviewer 8JaY (Part 2)**
> >
> > **Q2.**
> >
> > > Unclear writing and motivation. The motivation behind the work is not clearly articulated. While the authors discuss how human visual perception is compositional, it remains unclear why such a decomposition framework is needed in the context of LVLMs or what specific applications it enables. Additionally, the stated contributions are somewhat confusing. One claim suggests that CRD improves the visual representation capabilities of LVLMs, while another claims that CRD outperforms standard LVLMs. Since CRD itself builds upon LVLMs, it is unclear whether the comparison is against the same backbone or a new model.
> >
> > We sincerely thank the reviewer for this feedback and apologize for the lack of clarity in our initial writing. Our goal is to enable LVLMs to reason about images in a more structured and interpretable manner by separating high-level concepts from their underlying visual rules, which supports downstream tasks such as meta-attribute extraction, abstract visual reasoning, and spatial relational reasoning. We expect  to clarify the contributions to avoid confusion. CRD does not create a new model; it builds directly on top of an existing LVLM and uses its ability to extract concepts and rules. Comparisons are made using the same LVLM backbone, without any additional training or parameter updates. In this sense, CRD does not replace the LVLM but instead reveals and improves its potential capabilities, which explains why it outperforms direct LVLM response while still being based on the same underlying model.
> >
> > **Q3.**
> >
> > > Lack of qualitative and quantitative baselines. The paper provides limited qualitative examples, making it difficult to understand what CRD produces in practice. Moreover, several relevant baselines such as SpatialVLM and VisuoThink, which also address compositional or structured visual reasoning, are discussed but not directly compared experimentally. Including such comparisons would help contextualize the claimed improvements.
> >
> > We thank the reviewer for this helpful suggestion. To better contextualize what CRD produces in practice and how it compares to existing methods, we have expanded both our qualitative and quantitative evaluations in the revised manuscript. On the quantitative side, as shown in Table 3, we additionally evaluate CRD on the SpatialEval benchmark [1], which covers a broader range of spatial and compositional reasoning tasks, and include direct comparisons with SpatialVLM on this benchmark. The results show that incorporating CRD leads to  consistent improvements over the backbone LVLMs and yields better performance compared to SpatialVLM, highlighting the ability of our framework beyond the original RAVEN setting. On the qualitative side, we have also added more illustrative examples of CRD’s outputs in the Appendix E.
> >
> >
> >
> > **Table 3. Results on the SpatialEval benchmark.**  SpatialEval includes four spatial reasoning tasks (SpatialMap, MazeNav, SpatialGrid, SpatialReal). CRD_meta denotes only the meta-attributes extracted by CRD are used as prompts to guide the LVLM’s responses. CRD_full further incorporates both the meta-attributes and the patch scores as additional prompts for the LVLMs.
> >
> >
> > $$
> > \\begin{array}{l|c|c|c|c|c}
> > \\hline
> > \\textbf{Model} & \\textbf{SpatialMap} & \\textbf{MazeNav} & \\textbf{SpatialGrid} & \\textbf{SpatialReal} & \\textbf{Overall} \\\\
> > \\hline
> > SpatialVLM                        & 48.87 & 22.60 & \\textbf{87.03} & 34.81 & 50.27 \\\\
> > Qwen2.5VL-3B                     & 50.93 & 27.40 & 83.33 & 91.85 & 54.00 \\\\
> > Qwen2.5VL-3B+CRD\\_meta          & 50.93 & \\textbf{29.07} & 85.20 & \\textbf{96.30} & 56.27 \\\\
> > Qwen2.5VL-3B+CRD\\_full          & \\textbf{51.67} & 29.00 & 86.40 & \\textbf{96.30} & \\textbf{56.87} \\\\
> > \\hline
> > Qwen2.5VL-7B                     & 63.00 & 28.93 & 85.60 & 91.11 & 60.11 \\\\
> > Qwen2.5VL-7B+CRD\\_meta          & 64.20 & 29.13 & 87.93 & 95.56 & 61.45 \\\\
> > Qwen2.5VL-7B+CRD\\_full          & \\textbf{65.26} & \\textbf{33.33} & \\textbf{88.46} & \\textbf{97.04} & \\textbf{63.37} \\\\
> > \\hline
> > \\end{array}
> > $$
> >
> >
> >
> > [1] Wang, Jiayu, et al. "Is a picture worth a thousand words? delving into spatial reasoning for vision language models." NeurIPS 2024.
> >
> > **Q4.**
> >
> > > Minor: The font size in Figure 1 is small and difficult to read; increasing it would improve presentation clarity.
> >
> > We thank the reviewer for pointing this out. We have increased the font size in Figure 1 and adjusted the layout in the revised manuscript to ensure better readability and overall presentation clarity.

---

### Official Review · Reviewer_1FDn · 2025-11-03

**Soundness:** 3
**Presentation:** 3
**Contribution:** 3
**Rating:** 6
**Confidence:** 3

**Summary:**

The paper proposes a Concept and Rule Decomposition (CRD) framework that enhances LVLMs with interpretable reasoning. CRD extracts visual concepts from images using an LVLM, models their spatial patterns via Gaussian Process–based Concept Rule Functions (CRFs), and selects the most consistent concept–rule pairs through LVLM-guided sampling. Experiments on real-image and abstract reasoning benchmarks show that CRD improves both performance and interpretability across diverse LVLMs.

**Strengths:**

1. It introduces a unified, model-agnostic approach that explicitly decomposes visual understanding into concepts and spatial rules, improving interpretability.
2. The proposed method consistently enhances multiple LVLMs across both real-image and abstract reasoning benchmarks without additional supervision.
3. The LVLM-guided sampling and GP-based rule modeling allow CRD to adapt flexibly to different models and datasets.

**Weaknesses:**

1. The method focuses mainly on spatial rules within static images and does not address temporal or relational reasoning across scenes.
2. The performance and interpretability depend heavily on the LVLM’s initial concept extraction accuracy. It would be better if there is human evaluation of LVLM’s initial concept extraction accuracy.
3. The Gaussian Process–based CRF and sampling procedure introduce extra computational overhead compared to direct LVLM inference.

**Questions:**

How sensitive is the CRD framework to the quality of concept proposals from different LVLMs, and could errors in these initial concepts propagate to rule learning and sampling outcomes?

---

> ### Author Response · Authors · 2025-11-27
> **Response to Reviewer 1FDn (Part 1)**
>
> We sincerely thank the reviewer for thoughtful evaluation and valuable feedback, which we have carefully considered and addressed in the responses below. And if you have any further questions, please feel free to post them, and we would be glad to address them.
>
> **Q1.**
>
> > The method focuses mainly on spatial rules within static images and does not address temporal or relational reasoning across scenes.
>
> We appreciate the reviewer’s suggestion. We have additionally evaluated CRD on the SpatialEval benchmark [1], which includes a broader range of relational reasoning tasks. As shown in Table 1, the improvements on the SpatialEval benchmark indicate that CRD can be effectively extended beyond the original setting, enhancing the model’s overall reasoning and understanding capabilities.
>
> **Table 1. Results on the SpatialEval benchmark.**  SpatialEval includes four spatial reasoning tasks (SpatialMap, MazeNav, SpatialGrid, SpatialReal). CRD_meta denotes only the meta-attributes extracted by CRD are used as prompts to guide the LVLM’s responses. CRD_full further incorporates both the meta-attributes and the patch scores as additional prompts for the LVLMs.
>
> $$
> \\begin{array}{l|c|c|c|c|c}
> \\hline
> \\textbf{Model} & \\textbf{SpatialMap} & \\textbf{MazeNav} & \\textbf{SpatialGrid} & \\textbf{SpatialReal} & \\textbf{Overall} \\\\
> \\hline
> SpatialVLM                        & 48.87 & 22.60 & \\textbf{87.03} & 34.81 & 50.27 \\\\
> Qwen2.5VL-3B                     & 50.93 & 27.40 & 83.33 & 91.85 & 54.00 \\\\
> Qwen2.5VL-3B+CRD\\_meta          & 50.93 & \\textbf{29.07} & 85.20 & \\textbf{96.30} & 56.27 \\\\
> Qwen2.5VL-3B+CRD\\_full          & \\textbf{51.67} & 29.00 & 86.40 & \\textbf{96.30} & \\textbf{56.87} \\\\
> \\hline
> Qwen2.5VL-7B                     & 63.00 & 28.93 & 85.60 & 91.11 & 60.11 \\\\
> Qwen2.5VL-7B+CRD\\_meta          & 64.20 & 29.13 & 87.93 & 95.56 & 61.45 \\\\
> Qwen2.5VL-7B+CRD\\_full          & \\textbf{65.26} & \\textbf{33.33} & \\textbf{88.46} & \\textbf{97.04} & \\textbf{63.37} \\\\
> \\hline
> \\end{array}
> $$
>
>
> [1] Wang, Jiayu, et al. "Is a picture worth a thousand words? delving into spatial reasoning for vision language models." NeurIPS 2024.

---

> > ### Author Response · Authors · 2025-11-27
> > **Response to Reviewer 1FDn (Part 2)**
> >
> > **Q2.**
> >
> > > The performance and interpretability depend heavily on the LVLM’s initial concept extraction accuracy.
> >
> > We thank the reviewer for raising this important point. In the revised version, we report the performance of multiple LVLM baselines on the full V* dataset. We also include their results on abstract visual reasoning tasks.
> >
> > **Table 2. Performance comparison on the full dataset.**
> >
> >
> > $$
> > \\begin{array}{l|c|c|c|c|c|c}
> > \\hline
> > \\textbf{Model} & \\textbf{Average Similarity} & \\textbf{Precision} & \\textbf{Recall} & \\textbf{F1} & \\textbf{AUPRC} & \\textbf{ROC-AUC} \\\\
> > \\hline
> >  InternVL3.5-4B        & 38.5 & 75.1 & 35.3 & 48.03 & 48.8 & 68.7 \\\\
> >  +CRD                  & 44.5 & 76.4 & 42.7 & 54.78 & 52.4 & 70.1 \\\\
> >  InternVL3.5-8B        & 59.9 & 75.7 & 51.2 & 61.08 & 65.2 & 83.9 \\\\
> >  +CRD                  & \\textbf{64.0} & \\textbf{77.4} & \\textbf{55.6} & \\textbf{64.71} & \\textbf{68.3} & \\textbf{84.8} \\\\
> > \\hline
> >  Qwen2.5VL-3B         & 31.5 & 77.1 & 26.9 & 39.88 & 42.5 & 65.8 \\\\
> >  +CRD                  & 36.7 & 77.3 & 32.8 & 46.06 & 47.9 & 68.1 \\\\
> > Qwen2.5VL-7B         & 46.9 & 73.7 & 38.0 & 50.15 & 54.1 & 74.6 \\\\
> > +CRD                  & \\textbf{51.6} & \\textbf{76.3} & \\textbf{44.4} & \\textbf{56.13} & \\textbf{58.0} & \\textbf{75.7} \\\\
> > \\hline
> >  DeepSeekVL2-Tiny     & 16.8 & 39.1 & 21.1 & 27.41 & 36.3 & 50.8 \\\\
> >   +CRD                  & \\textbf{20.4} & \\textbf{44.8} & \\textbf{23.2} & \\textbf{30.57} & \\textbf{40.1} & \\textbf{58.3} \\\\
> > \\hline
> > \\end{array}
> > $$
> >
> >
> >
> >
> > Our motivation is to decompose concepts and rules based on the LVLM’s existing capabilities, so the quality of the underlying model naturally affects CRD’s outputs. If an LVLM has no meaningful ability to extract or distinguish concepts at all, CRD can hardly create improvements from scratch. Our results across different baselines show that as long as the LVLM possesses a reasonable level of initial concept extraction ability, CRD consistently enhances its performance. Table 2 demonstrates the performance gains brought by CRD across different backbones in meta-attribute extraction task. As shown in Tables 3 and 4, Qwen also achieves substantial gains on RAVEN and I-RAVEN after applying CRD. Since all tested models exhibit at least initial capability, our method reliably improves their outcomes.
> >
> > **Table 3. Performance comparison on the RAVEN dataset.**
> >
> >
> > $$
> > \\begin{array}{l|c|c|c|c|c|c|c|c}
> > \\hline
> > \\textbf{Model} & \\textbf{Average} & \\textbf{CS} & \\textbf{LR} & \\textbf{UD} & \\textbf{OIC} & \\textbf{OIG} & \\textbf{2Grid} & \\textbf{3Grid} \\\\
> > \\hline
> > Qwen2.5VL-7B          & 59.7 & \\textbf{78.0} & 53.0 & 55.0 & 73.0 & 65.0 & 52.0 & 42.0 \\\\
> > Qwen2.5VL-7B+CRD      & \\textbf{89.4} & 77.0 & \\textbf{97.0} & \\textbf{95.0} & \\textbf{87.0} & \\textbf{98.0} & \\textbf{84.0} & \\textbf{88.0} \\\\
> > \\hline
> > InternVL3.5-8B        & 11.6 & 16.0 & 6.0  & 12.0 & 11.0 & 17.0 & 13.0 & 6.0  \\\\
> > InternVL3.5-8B+CRD    & \\textbf{31.6} & \\textbf{22.0} & \\textbf{19.0} & \\textbf{31.0} & \\textbf{33.0} & \\textbf{47.0} & \\textbf{33.0} & \\textbf{36.0} \\\\
> > \\hline
> > \\end{array}
> > $$
> >
> >
> >
> > **Table 4. Performance comparison on the I-RAVEN dataset.**
> >
> >
> > $$
> > \\begin{array}{l|c|c|c|c|c|c|c|c}
> > \\hline
> > \\textbf{Model} & \\textbf{Average} & \\textbf{CS} & \\textbf{LR} & \\textbf{UD} & \\textbf{OIC} & \\textbf{OIG} & \\textbf{2Grid} & \\textbf{3Grid} \\\\
> > \\hline
> > Qwen2.5VL-7B        & 15.0 & 19.0 & 18.0 & 14.0 & 18.0 & 12.0 & 9.0  & 15.0 \\\\
> > Qwen2.5VL-7B+CRD    & \\textbf{89.3} & \\textbf{81.0} & \\textbf{93.0} & \\textbf{95.0} & \\textbf{89.0} & \\textbf{98.0} & \\textbf{80.0} & \\textbf{89.0} \\\\
> > \\hline
> > InternVL3.5-8B      & 13.9 & 15.0 & 14.0 & 13.0 & 13.0 & 14.0 & 11.0 & 17.0 \\\\
> > InternVL3.5-8B+CRD  & \\textbf{33.6} & \\textbf{24.0} & \\textbf{29.0} & \\textbf{37.0} & \\textbf{44.0} & \\textbf{49.0} & \\textbf{20.0} & \\textbf{32.0} \\\\
> > \\hline
> > \\end{array}
> > $$

---

> > > ### Author Response · Authors · 2025-11-27
> > > **Response to Reviewer 1FDn (Part 3)**
> > >
> > > **Q3.**
> > >
> > > > It would be better if there is human evaluation of LVLM’s initial concept extraction accuracy.
> > >
> > > In addition, we collected concept annotations produced solely by human experts and compared model predictions against these purely human-labeled references. As shown in Table 5, the similarity scores and other evaluation metrics exhibit the same performance trends as those obtained using our original set (human experts + GPT annotations), further confirming the reliability and effectiveness of our method.
> > >
> > > **Table 5. Performance comparison on the full dataset with human annotation.**
> > >
> > >
> > > $$
> > > \\begin{array}{l|c|c|c|c|c|c}
> > > \\hline
> > > \\textbf{Model} & \\textbf{Average Similarity \\S} & \\textbf{Precision \\S} & \\textbf{Recall \\S} & \\textbf{F1 Score \\S} & \\textbf{AUPRC \\S} & \\textbf{ROC-AUC \\S} \\\\
> > > \\hline
> > > InternVL3.5-4B        & 50.36 & 66.72 & 40.21 & 50.18 & 68.33 & 78.10 \\\\
> > > +CRD                  & 58.30 & \\textbf{82.15} & 47.76 & 60.40 & 72.06 & 82.85 \\\\
> > > InternVL3.5-8B        & 58.56 & 72.31 & 51.39 & 60.08 & 72.71 & 89.51 \\\\
> > > +CRD                  & \\textbf{69.22} & 80.02 & \\textbf{60.81} & \\textbf{69.10} & \\textbf{79.13} & \\textbf{92.17} \\\\
> > > \\hline
> > > Qwen2.5-VL-3B         & 43.81 & 60.37 & 33.31 & 42.93 & 65.63 & 76.47 \\\\
> > > +CRD                  & 49.40 & 73.92 & 37.68 & 49.92 & 69.58 & 82.24 \\\\
> > > Qwen2.5-VL-7B         & 55.27 & 64.03 & 47.21 & 54.35 & 69.87 & 82.74 \\\\
> > > +CRD                  & \\textbf{61.86} & \\textbf{76.19} & \\textbf{52.50} & \\textbf{62.16}  & \\textbf{73.67} & \\textbf{87.31} \\\\
> > > \\hline
> > > DeepSeek-VL2-Tiny     & 25.72 & 31.00 & 23.37 & 26.65 & 55.04 & 66.19 \\\\
> > > +CRD                  & \\textbf{29.01} & \\textbf{40.25} & \\textbf{26.16} & \\textbf{31.71} & \\textbf{57.12} & \\textbf{69.64} \\\\
> > > \\hline
> > > \\end{array}
> > > $$
> > >
> > >
> > > **Q4.**
> > >
> > > > The Gaussian Process–based CRF and sampling procedure introduce extra computational overhead compared to direct LVLM inference.
> > >
> > > We appreciate the reviewer’s concern about computational overhead. In the revised manuscript, we include a table summarizing the inference efficiency of different models with and without CRD. As shown in Table 6, the GP-based CRF and sampling procedure do introduce additional computation compared to direct LVLM inference. However, this extra cost comes with more accurate concept-rule decomposition, which in turn leads to significantly improved performance on the evaluated reasoning tasks.
> > >
> > > **Table 6. Efficiency Analysis Table.**
> > >
> > >
> > > $$
> > > \\begin{array}{l|c|c|c|c|c}
> > >  \\textbf{Model} & \\textbf{Latency Per Token (ms)} & \\textbf{TFLOPs} & \\textbf{GPU Memory (GiB)} & \\textbf{KV-Cache (MB)} & \\textbf{Input Token} \\\\
> > > \\hline
> > >   InternVL3.5-8B        & 77.3  & 39.02 & 17.02 & 1287.0 & 2288 \\\\
> > >   +CRD                  & 234.3 & 42.93 & 17.46 & 1405.7 & 2499 \\\\
> > > \\hline
> > >   Qwen2.5VL-7B          & 39.7  & 6.47  & 16.49 & 171.1  & 447  \\\\
> > >   +CRD                  & 190.1 & 8.91  & 17.32 & 234.7  & 613  \\\\
> > > \\hline
> > >   DeepSeekVL2-Tiny      & 59.6  & 1.06  & 11.82 & 114.8  & 1224 \\\\
> > >   +CRD                  & 264.1 & 1.16  & 12.53 & 124.9  & 1332 \\\\
> > > \\hline
> > > \\end{array}
> > > $$
> > >
> > >
> > > **Q5.**
> > >
> > > > How sensitive is the CRD framework to the quality of concept proposals from different LVLMs, and could errors in these initial concepts propagate to rule learning and sampling outcomes?
> > >
> > > As shown in Table 2, different LVLM baselines indeed start with varying levels of concept proposal quality, and this naturally influences the downstream rule learning and sampling results. The abstract reasoning experiments (Tables 3 and 4) further illustrate how CRD behaves across models with different initial abilities. While errors in the initial concepts can propagate to some extent, CRD improves performance by leveraging the useful structure in the LVLM. In other words, CRD unlocks more of the model’s potential, and the stronger the underlying LVLM, the greater the performance gain we observe.

---

### Official Review · Reviewer_EaHP · 2025-11-03

**Soundness:** 2
**Presentation:** 1
**Contribution:** 2
**Rating:** 2
**Confidence:** 2

**Summary:**

The paper describes a procedure for concept extraction that sits on top of generic LVLMs. The method samples a set of "visual concepts" that appear in the image, and the learns a set of "rules" which relate them--where the rules appear to be spatial positions of the identified concepts (this is not entirely clear to me from reading, see questions below). The model is evaluated on two tasks, the first is a visual attribute extraction task (something like high level object recognition) and the second is RAVEN. The model performs better than generic LVLMs in both cases.

I am lukewarm about this paper because I think I am missing some key aspects of the intuition and motivation behind the model. The paper is a bit hard to read -- it is heavy on notation and formalization but lacking in intuition and examples. I thus don't feel I am fully able to appreciate what the contribution is, and how significant it would be. If the authors can provide more clarification in their response, I would feel more comfortable making a stronger recommendation.

**Strengths:**

* The paper presents a new procedure for making LLMs perform better on some visual reasoning tasks

**Weaknesses:**

* As written, the exact contribution and significance thereof is not very clear. Please see my questions below. Without better explanation and motivation, its hard to tell if how impactful the work can potentially be

**Questions:**

* My understanding is that "concepts" you extract are just objects, and the "rules" you extract are just estimates of the possible positions (spatial location) of detected objects. Is this correct? If not, can you give an example of a concept that is extracted that is not a noun/object and a rule that isn't spatial? I ask because this greatly affects the generalizability of the architecture -- many concepts don't readily fit this framework. E.g., the concept of "wearing" itself refers to a spatial relationship between objects, so is higher order than what (I think) your method can handle. Apologies if I misunderstood your method, please correct me if I did.
* Can you explain what role the "rules" play in the meta-attribute extraction task? I don't understand the motivation for this task -- it looks like it is just object recognition (with some higher-level labels). Why is this task a good illustration of the strengths of your method?
* Also on the attribute extraction task -- why did you only select 100 examples? Why could you not just use the whole dataset for evaluation?
* You place emphasis on the generality of your procedure, as it is based on LVLMs which are generic. However, you don't actually demonstrate that a system that has undergone CRD training is still generic -- i.e., does it retain the general purpose abilities that it had before the CRD was applied? The results give the feeling that it is in fact a procedure that is fairly specific to RAVEN, and the results are primarily on RAVEN. It would be good to demonstrate that this is not the case by evaluating on more diverse types of tasks to back of the claim of genericity. E.g., in the intro, you mention several probabilistic programming models such as Lake et al...would your procedure extend to those tasks?
* Can you say how many parameters are involved in fitting this model? Apologies if I missed this, but it wasn't clear to me how much to consider this a new model vs. a wrapper on a pretrained model

Typos: check your citation formats to make sure they are standard. E.g., when the citation is inline with the text, it should be in caps (e.g., "Jane Doe (2015) previously showed this was possible"), and otherwise in parenthesis (e.g. "It has been previously shown that this is possible (Doe, 2025)")

---

> ### Author Response · Authors · 2025-11-27
> **Response to Reviewer EaHP (Part 1)**
>
> We sincerely thank the reviewer for the constructive comments. We address the concerns as follows. If you have any further questions or suggestions, we would sincerely welcome the opportunity to address them.
>
> **Q1.**
>
> >My understanding is that "concepts" you extract are just objects, and the "rules" you extract are just estimates of the possible positions (spatial location) of detected objects. Is this correct? If not, can you give an example of a concept that is extracted that is not a noun/object and a rule that isn't spatial? I ask because this greatly affects the generalizability of the architecture -- many concepts don't readily fit this framework. E.g., the concept of "wearing" itself refers to a spatial relationship between objects, so is higher order than what (I think) your method can handle. Apologies if I misunderstood your method, please correct me if I did.
>
> We thank the reviewer for the insightful comments. Our definition of visual concepts (meta-attributes) is not limited to concrete objects. Instead, concepts refer to high-level, interpretable semantic properties that describe what is present or implied in an image. This includes relational or functional attributes such as wearing, which the reviewer mentioned. In the revised Appendix E, we illustrated how the wearing concept is grounded in the image. For this concept, LVLM produces a score over all image patches, where the score on each patch reflects the strength of its relationship to the attribute.
>
> Beyond relational attributes, our method also extracts abstract, non-noun concepts from synthetic reasoning datasets, including shape, size, and other global visual traits describing the overall image composition, as shown in Appendix E. These results align with our definition that a visual concept (meta-attribute) represents high-level semantic properties rather than concrete objects.
>
> **Q2.**
>
> > Can you explain what role the "rules" play in the meta-attribute extraction task? I don't understand the motivation for this task -- it looks like it is just object recognition (with some higher-level labels). Why is this task a good illustration of the strengths of your method?
>
> The goal of the meta-attribute extraction task is to identify which high-level concepts best describe a given image, rather than to recognize objects per se. In this task, the “rules” model how each candidate meta-attribute is distributed over image patches, capturing where and how strongly the attribute is expressed in the scene. As illustrated by the wearing example in the revised Appendix E, when patches involve people wearing specific clothing or not wearing a top, the learned rule assigns higher scores to these regions, which in turn leads to a higher probabilities for the wearing attribute, indicating that it is a good descriptor of that image. We also report CRD scores and patch–attribute relevance maps for multiple meta-attributes that are either clearly related or unrelated to the image content, showing that the rules can discriminate relevant and spurious attributes. Therefore, this task provides a quantitative and interpretable demonstration of how our rule extraction mechanism helps select the most appropriate meta-attributes for describing an image.
>
> **Q3.**
>
> > Also on the attribute extraction task -- why did you only select 100 examples? Why could you not just use the whole dataset for evaluation?
>
> For the attribute extraction task, the experiments used a random subset of 100 images in order to accelerate validation and reduce computation. As shown in the table below, we have evaluated our method on the full dataset. The outcomes remain consistent with those obtained from the 100-sample subset.
>
> **Table 1. Performance comparison on the full dataset.**
>
>
> $$
> \\begin{array}{l|c|c|c|c|c|c}
> \\hline
> \\textbf{Model} & \\textbf{Average Similarity} & \\textbf{Precision} & \\textbf{Recall} & \\textbf{F1} & \\textbf{AUPRC} & \\textbf{ROC-AUC} \\\\
> \\hline
>  InternVL3.5-4B        & 38.5 & 75.1 & 35.3 & 48.03 & 48.8 & 68.7 \\\\
>  +CRD                  & 44.5 & 76.4 & 42.7 & 54.78 & 52.4 & 70.1 \\\\
>  InternVL3.5-8B        & 59.9 & 75.7 & 51.2 & 61.08 & 65.2 & 83.9 \\\\
>  +CRD                  & \\textbf{64.0} & \\textbf{77.4} & \\textbf{55.6} & \\textbf{64.71} & \\textbf{68.3} & \\textbf{84.8} \\\\
> \\hline
>  Qwen2.5VL-3B         & 31.5 & 77.1 & 26.9 & 39.88 & 42.5 & 65.8 \\\\
>  +CRD                  & 36.7 & 77.3 & 32.8 & 46.06 & 47.9 & 68.1 \\\\
> Qwen2.5VL-7B         & 46.9 & 73.7 & 38.0 & 50.15 & 54.1 & 74.6 \\\\
> +CRD                  & \\textbf{51.6} & \\textbf{76.3} & \\textbf{44.4} & \\textbf{56.13} & \\textbf{58.0} & \\textbf{75.7} \\\\
> \\hline
>  DeepSeekVL2-Tiny     & 16.8 & 39.1 & 21.1 & 27.41 & 36.3 & 50.8 \\\\
>   +CRD                  & \\textbf{20.4} & \\textbf{44.8} & \\textbf{23.2} & \\textbf{30.57} & \\textbf{40.1} & \\textbf{58.3} \\\\
> \\hline
> \\end{array}
> $$

---

> > ### Author Response · Authors · 2025-11-27
> > **Response to Reviewer EaHP (Part 2)**
> >
> > **Q4.**
> >
> > > You place emphasis on the generality of your procedure, as it is based on LVLMs which are generic. However, you don't actually demonstrate that a system that has undergone CRD training is still generic -- i.e., does it retain the general purpose abilities that it had before the CRD was applied? The results give the feeling that it is in fact a procedure that is fairly specific to RAVEN, and the results are primarily on RAVEN. It would be good to demonstrate that this is not the case by evaluating on more diverse types of tasks to back of the claim of genericity. E.g., in the intro, you mention several probabilistic programming models such as Lake et al...would your procedure extend to those tasks?
> >
> > Regarding the concern about genericity, CRD does not modify or fine-tune the underlying LVLM. It operates as an external decomposition module that leverages the LVLM’s pretrained representations to extract concepts and rules. These concepts and rules can further facilitate downstream tasks such as meta-attribute extraction and abstract visual reasoning. Since no parameters of the LVLM are updated, all original general-purpose abilities are fully preserved after applying CRD. To further demonstrate that the procedure is not specific to RAVEN, as shown in the table below, we additionally evaluated CRD on the SpatialEval benchmark, which involves a different set of visual-spatial reasoning tasks. The improvements support the claim that CRD is not tied to a particular dataset, e.g., RAVEN.
> >
> > **Table 2. Results on the SpatialEval benchmark.**  SpatialEval includes four spatial reasoning tasks (SpatialMap, MazeNav, SpatialGrid, SpatialReal). CRD_meta denotes only the meta-attributes extracted by CRD are used as prompts to guide the LVLM’s responses. CRD_full further incorporates both the meta-attributes and the patch scores as additional prompts for the LVLMs.
> >
> >
> > $$
> > \\begin{array}{l|c|c|c|c|c}
> > \\hline
> > \\textbf{Model} & \\textbf{SpatialMap} & \\textbf{MazeNav} & \\textbf{SpatialGrid} & \\textbf{SpatialReal} & \\textbf{Overall} \\\\
> > \\hline
> > SpatialVLM                        & 48.87 & 22.60 & \\textbf{87.03} & 34.81 & 50.27 \\\\
> > Qwen2.5VL-3B                     & 50.93 & 27.40 & 83.33 & 91.85 & 54.00 \\\\
> > Qwen2.5VL-3B+CRD\\_meta          & 50.93 & \\textbf{29.07} & 85.20 & \\textbf{96.30} & 56.27 \\\\
> > Qwen2.5VL-3B+CRD\\_full          & \\textbf{51.67} & 29.00 & 86.40 & \\textbf{96.30} & \\textbf{56.87} \\\\
> > \\hline
> > Qwen2.5VL-7B                     & 63.00 & 28.93 & 85.60 & 91.11 & 60.11 \\\\
> > Qwen2.5VL-7B+CRD\\_meta          & 64.20 & 29.13 & 87.93 & 95.56 & 61.45 \\\\
> > Qwen2.5VL-7B+CRD\\_full          & \\textbf{65.26} & \\textbf{33.33} & \\textbf{88.46} & \\textbf{97.04} & \\textbf{63.37} \\\\
> > \\hline
> > \\end{array}
> > $$
> >
> >
> > [1] Wang, Jiayu, et al. "Is a picture worth a thousand words? delving into spatial reasoning for vision language models." NeurIPS 2024.
> >
> > **Q5.**
> >
> > > Can you say how many parameters are involved in fitting this model? Apologies if I missed this, but it wasn't clear to me how much to consider this a new model vs. a wrapper on a pretrained model
> >
> > For clarity, CRD is implemented as an independent module rather than a new model that modifies or fine-tunes the LVLM. The CRD module consists of only a few lightweight neural network layers and introduces merely 0.00001B trainable parameters. The pretrained LVLM remains entirely frozen. Therefore, CRD can be plugged into LVLM without altering its original weights or compromising its general-purpose abilities.
> >
> > **Q6.**
> >
> > > Typos: check your citation formats to make sure they are standard. E.g., when the citation is inline with the text, it should be in caps (e.g., "Jane Doe (2015) previously showed this was possible"), and otherwise in parenthesis (e.g. "It has been previously shown that this is possible (Doe, 2025)")
> >
> > We thank the reviewer for pointing out the issues in our citation formatting. We have carefully checked and correct the typos, and have incorporated these fixes into the revised version.

---

### Author Response · Authors · 2025-12-01
**Rebuttal Summary**

We sincerely thank the reviewers and the AC for their thoughtful feedback and careful evaluation. The detailed comments greatly helped us strengthen the clarity, completeness, and empirical depth of the paper. Many of the insights raised in the reviews directly motivated substantial improvements in both writing and experiments.

To address the reviewers’ concerns comprehensively, we implemented a series of revisions and additions across the manuscript:

1. **Clarification of key definitions.** We refined the explanation of *visual concepts* (meta-attributes) and *rules*, emphasizing that concepts are not restricted to objects but include high-level semantic, relational, and functional properties such as *wearing*. We also clarified how rules describe the distribution strength and spatial expression of each concept across image patches (EaHP, 8JaY).
2. **Clarification of the CRD framework.** We emphasized that CRD is a plug-in framework that does not alter the LVLM architecture or parameters; it leverages the model’s inherent conceptual understanding to decompose images into interpretable concepts and rules (EaHP, 8JaY).
3. **Clarification of human annotation.** We revised the description of human expert annotations and added the corresponding details in Appendix F, addressing concerns about annotation procedure and reliability (3kLB).
4. **Full-dataset evaluation.** We added experiments on the full V* dataset to demonstrate the robustness of CRD, now reflected in Table 1 of the manuscript, directly responding to questions about training and evaluation scale (EaHP, 3kLB).
5. **Human evaluation results.** To validate the reliability of our metrics, we added human-aligned evaluation results. These results match our automatic assessments, as shown in Table 1 and Table 4 of the manuscript, supporting that CRD’s improvements are also recognized under human judgment (1FDn, 8JaY).
6. **Additional spatial-reasoning benchmark.** We included experiments on an extra spatial-reasoning benchmark to demonstrate the extensibility of CRD beyond abstract reasoning, reported in Table 5 of the manuscript. This shows that CRD can generalize to broader spatial reasoning scenarios (EaHP, 1FDn, 3kLB).
7. **Expanded baseline comparisons.** We incorporated more specialized baselines (SpatialVLM, ResNet+DRT, LEN, SRAN) to more thoroughly evaluate the effectiveness of CRD, shown in Table 3 and Table 5 of the manuscript. This directly addresses requests for comparisons with stronger and task-specific methods (8JaY, 3kLB).
8. **Efficiency analysis.** We conducted an in-depth analysis of CRD’s runtime and the effect of patch count on efficiency. Results in Table 6 and Table 7 of the manuscript show that the additional computation is small relative to the performance gain, and patch count does not significantly affect runtime, clarifying the practical overhead of CRD in deployment (1FDn, 3kLB).
9. **Model-sensitivity study.** We evaluated CRD on InternVL3, Qwen2.5VL, and DeepSeekVL2 across multiple benchmarks to show how stronger base models benefit more from CRD. These results, in Table 1, Table 3, and Table 4 of the manuscript, illustrate that CRD can unlock the conceptual reasoning potential of different LVLMs and is not tied to a single backbone (1FDn, 3kLB).
10. **Additional visualizations.** We added more visual case studies to illustrate concept extraction, rule patterns, and the interpretability advantages of CRD, now included as Figures 9–11, making the qualitative behavior of CRD more transparent (EaHP, 8JaY).

We appreciate the reviewers’ insightful suggestions and the AC’s effort in the review process. The feedback significantly improved the presentation and empirical rigor of our work. We hope the revised manuscript now effectively resolves the reviewers' concerns.

---

### Meta-Review · Area_Chair_xbji · 2025-12-29

**Summary:**

This paper proposed Concept and Rule Decomposition (CRD), a framework that aims to enable LVLMs to discover concept-rule pairs from images.

After the review stage, a reviewer suggested to reject (although with a low confidence, i.e. 2), while the other three are on a borderline, with two borderline rejects and one borderline accept (4,4,6). Overall, the paper was on the edge, with a negative outlook.

The AC interpreted some of the criticisms from the reviewers to be of limited relevance for the final decision, such as the fact the work does not take into account the temporal dimension or that the method requires extra compute.
In the AC's opinion, some main aspects brought up by the reviewers were
* need for more qualitative/quantitative evidence for the interpretability claims
* missing of relevant baselines
* potential lack of generalizability (due to small scale of reported results)

The authors provided a rich rebuttal, which tried to touch on most of these aspects (see also additional AC comments under "Reviewer Concerns").

Overall, the AC's opinion is that the main concerns have been addressed through the rebuttal. As highlighted by the rebuttal, the paper needed (and still partially needs) to undergo a major review to incorporate all the suggested changes and gaps highlighted by the reviewers. Nevertheless the novelty of the work and the quality of the reported results make, in my opinion, the needle lean more on acceptance. Assuming in good faith that the authors will be able to appropriately incorporate these changes into the paper, the AC's recommendation leans towards accept.

**Reviewer Concerns:**

During the rebuttal, the authors provided a large set of changes and improvements (see summary list in 10 points), which includes various clarifications, additional evaluations and benchmarks, additional analysis, comparisons and visualizations.

In particular the authors provided additional results on the SpatialEval benchmark and including more baselines. The encouraging results reported seem to address the concerns from the reviewers about the need to compare against certain baselines (i.e. LEN, ResNet + DRT, SRAN) and the need for more quantitative evaluation.

The comment on the generalizability was address through the following comment:
"we used a randomly selected 100-image subset to accelerate validation and analysis. In the revised manuscript, we report results on the full V* dataset to address concerns about sample size and selection, and the conclusions remain consistent with those from the subset."
This and the additional evaluation on SpatialEval seem to address this concern.

**Reviewer Scores:**

While it is hard to predict how the reviewers would have changed their scores, I could assume that one or more of the borderline reviewers would have considered raising their scores, bringing the average to more positive territory.

---

### Decision · Program_Chairs · 2026-01-26

Accept (Poster)